# Improving Attributed Long-form Question Answering with Intent Awareness

**Xinran Zhao**[1,2*], **Aakanksha Naik**[1], **Jay DeYoung**[1], **Joseph Chee Chang**[1],

**Jena D. Hwang**[1], **Tongshuang Wu**[2], **Varsha Kishore**[1,3]

[1]Allen Institute for AI,  [2]Carnegie Mellon University,  [3]University of Washington

## Abstract

Large language models (LLMs) are increasingly being used to generate comprehensive, knowledge-intensive reports. However, while these models are trained on diverse academic papers and reports, they are not exposed to the reasoning processes and intents that guide authors in crafting these documents. We hypothesize that enhancing a model's intent awareness can significantly improve the quality of generated long-form reports. We develop and employ a structured tag-based schema to elicit underlying intents more effectively for writing. We demonstrate that these extracted intents enhance both zero-shot generation capabilities in LLMs and enable the creation of high-quality synthetic data for fine-tuning smaller models. Across various challenging scientific report generation tasks, our experiments show average improvements of +2.9 and +12.3 absolute points for large and small models over baselines, respectively. Furthermore, our analysis illuminates how intent awareness enhances model citation usage and substantially improves report readability.

## 1 Introduction

Recent advances in LLMs have fueled growing interest in building deep research systems that analyze information from various sources and produce a detailed report (DeepMind, 2025; OpenAI, 2025; Skarlinski et al., 2024; Yang et al., 2024; Singh et al., 2025). Unlike older question answering systems, which focus on retrieving a few relevant documents to produce concise answers to specific questions, deep research systems aim to gather dozens or even hundreds of sources and organize them into a coherent report. These often lengthy reports demand more than simply aggregating retrieved content; they require careful organization of information, structured argumentation to weave evidence into a coherent narrative, and proper attribution of sources.

In this work, we argue that deep research systems can benefit from strategies humans use during the process of sensemaking (Pirolli & Card, 2005) and writing (Flower & Hayes, 1981). Humans write with *intent*—every paragraph and sentence serves a particular purpose (Lauscher et al., 2022). Much of this intent remains invisible in the final text, though its role is measurable through observation: a recent study recorded scholars writing on Overleaf (Wang et al., 2025) showed nearly 10% of keystrokes were devoted to outlining, planning, and organization. These high-level intents, though essential for guiding the writing process, are not preserved in the final written texts and thus remain absent from data used to train language models. Therefore, models learn to mimic human writing style but don't explicitly model the thought process that goes into writing.

To model the thought process, we explore whether incorporating intent awareness helps language models generate better quality text, especially for scientific deep research tasks. Specifically, we propose an intent-aware writing framework that consists of intents at two levels–paragraphs and citations. These intents are represented in a tag-based format inspired by STaR (Zelikman et al., 2022) and ToW (Xu et al., 2025), where we include a dedicated intent type and a natural language rationale (see Figure 1 for an example). For citation intents, we identify fine-grained intent types from

---

\* Correspondance to xinranz3@andrew.cmu.edu. Our code and data is at: `https://github.com/colinzhaoust/intent-aware-deep-research`.

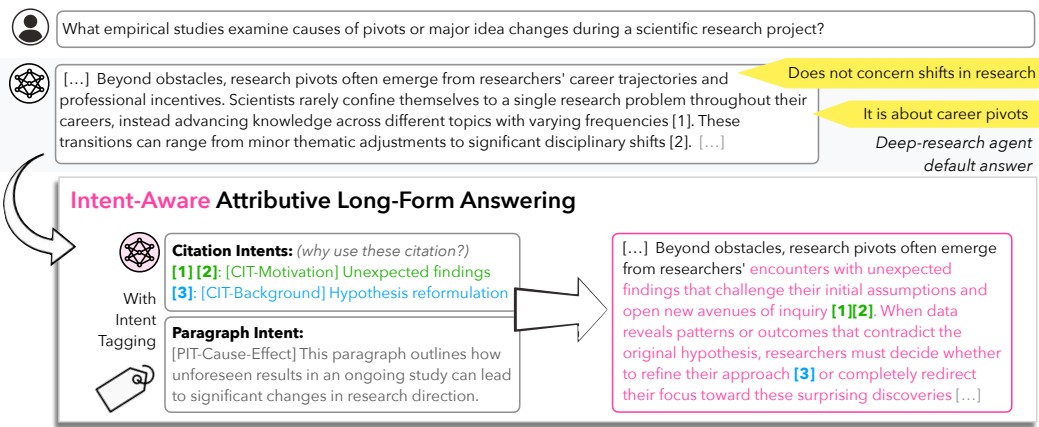

Figure 1: Current long-form question answering systems don't consider intents when generating responses. The figure above shows how having explicit citation intents and paragraph intents helps reason about the text and generate better responses.

literature on citation intent classification (Teufel et al., 2006; Jurgens et al., 2018; Cohan et al., 2019; Lauscher et al., 2022). Specifically, we adopt the six-category framework from ACL-ARC (Jurgens et al., 2018) in our pipeline, which includes categories such as *Background*, *Motivation*, and *Uses*. For paragraph intents, we choose fine-grained types from well-established discourse modes that capture the functional purpose of writing (Smith, 2003; Song et al., 2017), with categories such as *Exposition*, *Definition*, *Compare and Contrast*, *Problem-solution*, etc.

We explore the effectiveness of our intent-aware writing framework in improving the performance of scientific deep research systems (Singh et al., 2025) when incorporated during inference as well as training. At inference time, intent awareness is incorporated by prompting a model to produce reports with embedded intents (Tian et al., 2023). For intent-aware training, we first prompt a teacher model to produce reports with embedded intents; then this data, containing intent information, is used as high-quality training data for smaller models.

We conduct experiments on three recent long-form report generation benchmarks (Bragg et al., 2025; Patel et al., 2025; Yifei et al., 2025). Our results show that the intent awareness we induced consistently improves model performance across model backbones and tasks. With a macro-average across metrics, we observe an average improvement of +2.9 absolute points for large commercial language models, despite their already strong base model performance. For small models, we observe a +12.3 absolute point improvement after intent-aware SFT, where the best variants reach the level of performance of larger models such as `gemini-2.5-pro`. The performance improvements in automatic evaluations are driven by substantial gains in citation metrics which evaluate attribution quality. With intent awareness, we see +3.7 and +18.7 absolute point gains (averaged across models), for large commercial language models and small models, respectively.

Finally, we perform a detailed analysis of how the model's citation behavior changes with intent awareness, e.g., small models will make use of more retrieved information. We also conduct a user study showing that intents produced by our system aid in transparency and improved readability of long-form answers. For instance, they help guide the reader's attention, especially when the material is unfamiliar to them. Our approach leverages intent awareness to produce attributed responses that are more focused, reliable, and useful, particularly in scientific reporting.

## 2 RELATED WORK

**Attributed long-form generation.** Associating claims with evidence from identifiable sources plays a key role in measuring the faithfulness of model-generated text (Bohnet et al., 2023; Rashkin et al., 2023). Researchers have studied citation quality in scientific text generation (Funkquist et al., 2022), generative search engines (Liu et al., 2023), and Wikipedia-style document generation (Gao et al., 2023b). Prior work has predominantly used retrieval-augmented generation frameworks (RAG,

Lewis et al., 2020), wherein LLMs are trained to incorporate external or parametric knowledge sources and supporting documents while generating citations (Nakano et al., 2022; Menick et al., 2022; Gao et al., 2023a). More recently, the introduction of *deep research* systems (DeepMind, 2025; OpenAI, 2025) has led to improved performance on knowledge based reasoning-intensive short-form question-answering tasks (Mialon et al., 2023; Phan et al., 2025; Wei et al., 2025; Coelho et al., 2025) through active and strategic usage of retrieval. Such success motivates the community to further assess broader capabilities of these systems in answering long-form open-ended questions from real-world applications, such as advanced search engine use (Du et al., 2025), scientific QA systems (Bragg et al., 2025), and literature synthesis and comparison (Patel et al., 2025; Yifei et al., 2025).

In our work, we explore how introducing intent awareness during writing and citation selection can improve performance on attributed long-form report generation tasks. Our work focuses on changes through decoding time strategies and distillation, leaving existing RAG-style architectures intact, enabling potential plug-and-play generalization.

**Intents in writing.** Understanding how people determine what to write has interested researchers for decades. Prior work has examined intents underlying both citations (Goodwin, 1980; Teufel et al., 2006) and discourse structures (Bain, 1890; Smith, 2003; Song et al., 2017), often treating intent understanding as a classification or parsing task, e.g., citation intent classification (Cohan et al., 2019) and discourse parsing (Marcu, 2000; Feng & Hirst, 2012; Li et al., 2014). Advances in LLMs have opened up new opportunities to assist and automate writing (Lee et al., 2024; Shen et al., 2023), motivating researchers to incorporate intent understanding into generation (Padmakumar et al., 2025; Wang et al., 2025). Recently, Wu et al. (2025) discussed the LLM sensitivity of human intentions and incentives. In our work, we propose methods to add intent awareness during both training and inference, resulting in improved LLM text generation capabilities.

**Learning with rationales.** Besides drawing inspiration from human writing processes, our method connects to prior work on eliciting additional context or rationales to augment and improve model generation, including chain-of-thought (CoT) style inference (Wei et al., 2022a; Kojima et al., 2022), rationale bootstrapping training (Zelikman et al., 2022), text metadata conditioning (Gao et al., 2025), word-level reasoning (Xu et al., 2025), and confidence verbalization to improve calibration (Tian et al., 2023). From this perspective, our intent-aware method can be viewed as a form of generative planning, where models explicitly reason about the organization of their outputs to improve both quality and transparency.

## 3 TASK FORMULATION AND METHODOLOGY

We first briefly describe the formulation of the attributed long-form question-answering task. Then, we introduce our intent-aware writing framework, describing: (i) how intents are represented while writing, and (ii) the types of intent that can be produced. Finally, we discuss how we incorporate intent awareness into LLMs for attributed long-form question answering, during both inference and training stages.

### 3.1 TASK FORMULATION

We focus on the task of attributed long-form scientific question answering, which is formalized as follows: given a user query $q$, a system is required to generate a multi-section report $\mathcal{R}$, where each section consists of multiple paragraphs $\{p_1, p_2, ...\}$. Each paragraph $p_i$ contains sentences $s_{i1}, s_{i2}, ...$ with supporting citations $c_1, c_2, ... \in C$ added wherever external references are required. The list of potential texts to cite ($\mathcal{C}$) can either come from the model's parametric knowledge or from retrieved documents.

### 3.2 INTENT-AWARE WRITING FRAMEWORK

We propose an intent-aware writing framework that incorporates two broad categories of intents, often used in prior work: (i) paragraph-level writing intents (**paragraph intents**, hereafter), and (ii) sentence-level **citation intents**. Paragraph intents specify the purpose of every paragraph $p_i$ within

Table 1: The types and descriptions for our intent awareness schemes. We adopt the citation intent types from ACL-ARC (Jurgens et al., 2018) and extend the paragraph intent types from the discourse modes studied in (Song et al., 2017).

| | Intent Category & Type | Description |
|---|---|---|
| **Citation Intent** | Background | The citation provides relevant information for this domain |
| | Motivation | The citation illustrates the need for data, goals, methods, etc. |
| | Uses | The sentence uses data, methods, etc. from the citation |
| | Extension | The sentence extends the referenced work's data, methods, etc. of the citation |
| | Comparison or Contrast | The sentence expresses similarity/differences to the referenced work |
| | Future | The citation identifies the reference as a potential avenue for future work |
| **Paragraph Intent** | Exposition | Explains, clarifies, or provides background information on a topic |
| | Definition | Defines a key term, concept, or theory with necessary boundaries |
| | Argumentation | Presents a claim supported by evidence, logic, or reasoning |
| | Compare-contrast | Highlights similarities and/or differences between subjects or findings |
| | Cause-effect | Explains causal relationships between events or phenomena |
| | Problem-solution | Identifies a problem and proposes a solution or response |
| | Evaluation | Assesses strengths, weaknesses, or significance according to criteria |
| | Narration | Recounts a sequence of events or chronological processes |

the overall narrative of the report (e.g., *this paragraph provides background context* or *this paragraph compares two state-of-the-art methods*). Citation intents, which are more granular, are designed to capture why a certain citation $c_j$ is used to support a particular sentence $s_{ix}$ (e.g., *this sentence uses the method proposed in the citation* or *this sentence expresses similarities to or differences from the cited work*). By first generating such intents, we provide the model with cues helpful for the writing process. We prime the model to consider intents during text generation.

**Intent Representation.** To distinguish the intent text from report text, we use an inline tag-based schema with rationales to represent intents. More specifically, intents are represented using the following template: *<begin intent>* [*intent type*] *rationale <end intent>*. For begin and end intent tags, we use <bcit><ecit> and <bpit><epit> for citation and paragraph intents, respectively. Rationales for paragraph intents are brief textual explanations of why the paragraph fits the chosen type, based on its planned content and function within the report. For citation intents, rationales typically explain the connection between the sentence containing the citation and a brief summary of supporting evidence from the cited reference.

**Intent Types.** Within both paragraph and citation intents, we utilize a more fine-grained set of intent types; these are listed in Table 1. For citation intents, we use the categories defined in ACL-ARC (Jurgens et al., 2018), and for paragraph intents we use the types defined in the discourse modes from (Song et al., 2017)[1].

### 3.3 INTENT AWARENESS DURING INFERENCE

First, we explore the effectiveness of incorporating our intent-aware writing framework at inference time for attributed long-form report generation. Models are prompted to directly output reports with paragraph and citation intent tags embedded within them. For paragraph intents, the intent tags are placed before the text of each paragraph. For citation intents, intent tags are placed between the citing sentence and the inline citation. This intent-aware prompting strategy, which we will refer to as *verbalized intents*, can be considered a variant of test-time scaling. It focused on eliciting a specific category of thoughts (i.e., intents) alongside the reports.

### 3.4 INTENT AWARENESS DURING TRAINING

Besides inference-time augmentation, we explore strategies to incorporate intent awareness during training. This is especially useful for smaller models, which already lag behind larger ones (Asta Bench, Bragg et al., 2025) because they struggle more with the added complexity introduced by explicit intent elicitation during report generation.

---

[1]We focus on the non-psycho-lingual functional intents and remove the *emotion expressing* mode.

For intent-aware training, we first apply our intent-aware prompting strategy to a large teacher model to produce training data with embedded intent tags and rationales. We then conduct supervised fine-tuning (SFT) on this data, in the following settings:

- `intent-implicit` SFT: We remove the embedded intent tags and rationales before training the smaller models. While the SFT training data is generated in an intent-aware manner, the intent information is not explicitly present during training: the large teacher model generates intents when producing the training data, but the small student models learn only the direct report generation task, not intent generation.
- `intent-explicit` SFT: This variant retains the embedded intent tags and rationales. These explicit tags can potentially help smaller models understand how to better structure paragraphs and use citations. This setting is motivated by previous work that augments training data with explanations (Murty et al., 2020) and thoughts (Xu et al., 2025).
- `intent-multiview` SFT: Previous variants require small models to learn how to use both citation and paragraph intents simultaneously. To further reduce the instruction complexity of each data point during training, we decompose intent-aware generation into multiple sub-tasks, corresponding to overall intent categories. Following Liang et al. (2023), for each data point, we produce four instruction-report pairs: (1) an intent-explicit version (intent tags/rationales retained); (2) a paragraph-intent version with only paragraph intents retained in prompts/reports; (3) a citation-intent version with only citation intents retained in prompts/reports; (4) a no-intent version with tags/rationales removed and the prompt scrubbed of intent-related instructions. We train a model on all of the instruction-report pairs (4x the instances of teacher-generated reports).

We consider two baselines: (1) directly prompting models without additional training or intent-awareness, (2) fine-tuning models on reports generated for the same query subset from the same teacher model, but without intent awareness (baseline SFT).

## 4 EXPERIMENTS AND ANALYSIS

### 4.1 EXPERIMENTAL SETTING

We conduct experiments on several recent datasets for attributed long-form text generation. These tasks expect long report-style answers to open-ended questions. We run experiments with the following three datasets:

**SQA-CS-V2** (Bragg et al., 2025): AstaBench provides a suite of tasks to allow a holistic measure of agents for scientific research, including literature understanding, data analysis, paper search, coding, etc[2]. We evaluate on their report generation benchmark `AstaBench-ScholarQA-CS2`. For this benchmark, the task is to generate reports for complex scientific questions. Our main results are on the 100-sample test set, and our ablations are on the 100-sample validation set. Each generated report is evaluated based on four metrics: rubric-based evaluation (whether key points identified by human-verified rubrics are contained in the answer), answer precision (whether each paragraph of the answer is on-topic and addresses the question), citation precision (whether the cited source text supports the claim), and citation recall (whether each claim in the answer is well-supported by citations, if necessary). These metrics were scored using an LLM-judge pipeline with answer decomposition and atomic evaluation.

**DeepScholar Bench** (Patel et al., 2025) is a benchmark for generating related-work sections for recent arχiv papers. The task involves retrieving, synthesizing, and citing prior research. Generated reports are judged for nugget coverage (are essential facts found in the report; akin to Rubric measures), organization (structure and coherence of system answer), citation precision (paralleling SQA-CS-V2 citation precision), and claim coverage (assesses fraction of claims that are fully supported by cited sources). We elide the retrieval quality metrics as we use a fixed retrieval set for all experiments. We use the 63 papers from the official GitHub Repository as our dataset. The original task involves writing a related work section for a given paper title and abstract. Since this task is under-specified, we slightly modify the task setting and generate the related work section using the title and the sub-section headers in the ground truth related work section.

---

[2] https://asta.allen.ai/chat

Table 2: Performance comparison across various models on SQA-CS-V2. *Overall* denotes the macro-average of other sub-metics. **Bold** indicates the best-performing row for overall metrics. *+intent* denotes the use of our intent-aware-writing framework with both paragraph and citaiton intents.

| Method | SQA-CS-V2 | | | | |
|---|---|---|---|---|---|
| | Overall | Rubrics | Ans. P | Citation P | Citation R |
| o3 | 85.1 | 91.4 | 96.5 | 89.4 | 63.4 |
| + intent | **86.0** | 90.7 | 96.6 | 89.9 | 66.9 |
| gemini-2.5-pro | 88.1 | 82.6 | 94.1 | 93.2 | 82.4 |
| + intent | **89.7** | 82.6 | 94.5 | 95.7 | 86.1 |
| Claude opus-4 | 85.4 | 84.3 | 87.9 | 89.6 | 79.6 |
| + intent | **89.0** | 85.5 | 89.3 | 95.1 | 86.0 |

Table 3: Performance comparison on DeepScholar Bench and ReseachQA. RQA denotes ResearchQA. *Overall* denotes the macro-average of other sub-metics. **Bold** indicates the best-performing row for overall metrics.

| Method | DeepScholar Bench (DSB) | | | | | RQA |
|---|---|---|---|---|---|---|
| | Overall | Nug. Cov. | Org. | Cite-P | Claim Cov | Rubrics |
| o3 | **46.8** | 47.0 | 61.1 | 39.1 | 40.2 | 76.3 |
| + intent | 43.2 | 49.1 | 64.1 | 27.2 | 34.3 | **79.3** |
| gemini-2.5-pro | 54.8 | 49.0 | 63.1 | 53.0 | 54.2 | 71.9 |
| + intent | **57.8** | 49.0 | 58.0 | 61.1 | 63.3 | **74.0** |
| Claude opus-4 | 58.1 | 54.0 | 64.1 | 56.6 | 57.6 | 74.3 |
| + intent | **59.9** | 53.3 | 65.3 | 60.1 | 61.1 | **75.7** |

**ResearchQA** (Yifei et al., 2025) is a dataset of twenty thousand queries (3.7k test), answers, and rubrics derived from survey articles written by humans. Every ResearchQA question is paired with rubrics generated from the same survey article as the one used for generating the question. We use the ResearchQA questions with the subdomain: Artificial Intelligence (50 test questions). Following the official benchmark guidelines, we report the averaged rubric scores (RQA) to evaluate responses to ResearchQA questions. Since the original paper shows better results in a parametric setting without retrieval, we follow this setting and only use paragraph intents for this task.

For all tasks, we use the official implementations for evaluation. For retrieval, we use the publicly available Semantic Scholar keyword search API (Kinney et al., 2023) and Semantic Scholar snippet search API (Singh et al., 2025). The retrieved snippets are often overly lengthy, so we use an LLM to extract only the salient parts. We fix the retrieved information set for each query in order to control for retrieval quality, only measuring writing performance differences in our experimental settings.

We test the effectiveness of intent-aware inference with commercial large language models, including o3 from OpenAI (OpenAI, 2025), gemini-2.5-pro (Comanici et al., 2025), and claude-4.1-opus (Anthropic, 2025). For intent-aware training, we utilize 1,000 random-sampled queries from OpenScholar (Asai et al., 2024) and generate synthetic data with gemini-2.5-pro. We use qwen3-4B/8B (Yang et al., 2025) and llama3.1-8B (Grattafiori et al., 2024) as the base models for SFT training. We open-source both our training data and model checkpoints to support future research in this area.

We compare all the variants in Section 4.2 with a control on the training steps, i.e., even if we can reformat 4x multiview data points from a certain number of data points generated from the large models, we use 1/4 steps to allow fair comparison in terms of compute. We include further details of the inference, training, and evaluation setup in Appendix A.1.

## 4.2 Experimental results

**Eliciting intents at test time improves model performance.** We test the effectiveness of the intents by eliciting intents directly during inference (see appendix A.6 for the prompt). Table 2 and Table 3 show that using intents leads to improved overall performance for all model backbones, despite

default generation from these models being a strong baseline. From the specific metric scores, we observe that intents help models to perform better attribution compared to default report generation: both citation metrics improve: citation precision and citation recall increase by 5-7 absolute points for Claude. The rubric score and the answer precision score, which do not consider citation quality, remain the same because state-of-the-art LLMs are already highly capable of extracting key facts from retrieved information and ensuring that the presented information is topically relevant to the query. To further validate the performance of rows with small margins, we conduct a paired t-test to test the hypothesis that +intent is better than default inference for the Overall scores. For `gemini-2.5-pro`, the p-value is 0.013; For `o3`, the p-value is 0.072. The low p-value shows that our results are statistically significant if we set alpha =0.1.

Interestingly, during our experiments, we observed that `o3` has much worse citation behavior than other frontier LLMs, especially on citation recall. From a qualitative analysis of 20 claims from `o3`-generated answers, we observe that for nearly 60% cases, the claims contain additional information about a paper added from `o3`'s own memory, going beyond the specific snippets provided from that paper in context. Adding citation intents seems to have mixed effects on this behavior, improving citation quality on AstaBench-SQA-CSV2 while dropping citation quality on DeepScholar Bench. Given the atypical citation behavior, we report the `o3` performance of our intent-aware inference with paragraph intent only in Appendix A.7. It achieves achieves 49.3 points overall, achieving a 2.5 absolute point gain over the default inference.

**Intent-aware generations help smaller models.** We further explore the effectiveness of intent-aware training with SQA-CS-V2. Table 4 presents the performance of different language models trained with the SFT variants described in Section 3.4. We test all the intent-aware method variants by prompting the resulting models to generate intents during inference. We ablate training with intents, by using the default inference prompt without explicitly asking for intents (Appendix; Table 7).

As shown in Table 4, across various LLMs, intent-aware SFT variants show improved performance when compared to no training or baseline SFT, with +7.9, 22.8, 6.1 absolute points of improvement compared to the base models, for `qwen3-8b`, `llama3.1-8b`, and `qwen3-4b`, respectively. For 8B models, intent-multiview SFT consistently leads to the best performance, surpassing `gemini-2.5-pro`, showing benefit from SFT with data points decomposed into multiple subtasks. For `qwen3-4b`, intent-explicit SFT and intent-multiview SFT perform much better than intent-implicit SFT; validating our hypothesis that the retained intent tags and rationales can potentially serve as extra explanations and help small models to better understand how to structure paragraphs and citations. As with the larger models, our performance gains primarily come from improved attribution (citation precision and citation recall).

To further validate the generalizability of our SFT variants, we further report the performance on DeepScholar Bench of the `qwen3-8b` variants in Appendix A.7. The best performing intent-implicit variant achieves 60.3 overall, which is better than the best performance large models, i.e., `Claude opus-4`, Table 3.

**Intent awareness influences model citation usage.** In addition to the performance improvements on the metrics listed above, we conduct an analysis on `gemini-2.5-pro` and `qwen3-8b` to understand how intent awareness during inference and training shapes the model behavior. Figure 2 presents the change of (1) average portion of retrieved candidates used in the report and (2) average coverage score between citations of `qwen3-8b` variants and `gemini-2.5-pro`. Adding intents at inference time significantly increases the portion of retrieved candidates (e.g., relevant papers) used in the report generation, without precision loss, as shown previously in Table 2. The increased retrieved candidate usage without precision loss indicates that the model can appropriately use a larger set of snippets to support the various claims in the answer. Similarly, intent-aware training leads to much higher retrieved candidates usage compared to the base model or baseline SFT, which sheds light on the model behavior change beyond averaged performance.

We also examine the overlap between citations used in the small models and citations used in `gemini-2.5-pro`. We find that this coverage analysis shows a similar trend: after intent-aware SFT, small models use citations like large models and the overlap in citations between the small and large models is much larger. Again we see that inference-time *verbalized intents* also consistently offer extra gain on the SFT-ed models.

Table 4: SQA-CS-V2 Performance Across different base models and method variants. For each of the intent-aware method variants, the inference prompt explicitly asks the model to use intents.

| Base Model | Variant | Overall | Rubrics | Answer P | Citation P | Citation R |
|---|---|---|---|---|---|---|
| gemini-2.5-pro(ref) | - | 88.1 | 82.6 | 94.1 | 93.2 | 82.4 |
| qwen3-8b | no training | 80.7 | 82.1 | 90.4 | 83.2 | 66.9 |
| | baseline SFT | 83.2 | 78.7 | 94.3 | 85.8 | 73.9 |
| | intent-explicit SFT | 88.0 | 80.5 | 93.0 | 93.6 | 85.0 |
| | intent-implicit SFT | 87.1 | 78.9 | 94.0 | 92.5 | 82.9 |
| | intent-multiview SFT | **88.6** | 81.4 | 94.7 | 93.7 | 84.7 |
| llama3.1-8B | no training | 66.4 | 64.6 | 77.5 | 67.2 | 56.1 |
| | baseline SFT | 84.4 | 78.1 | 92.3 | 89.8 | 77.4 |
| | intent-explicit SFT | 85.8 | 77.6 | 93.1 | 90.5 | 82.2 |
| | intent-implicit SFT | 87.8 | 77.9 | 93.3 | 94.0 | 85.9 |
| | intent-multiview SFT | **89.2** | 79.5 | 95.1 | 95.4 | 86.7 |
| qwen3-4b | no training | 80.9 | 78.0 | 94.6 | 82.8 | 68.1 |
| | baseline SFT | 83.4 | 80.1 | 92.4 | 86.2 | 74.8 |
| | intent-explicit SFT | **87.5** | 80.1 | 97.0 | 91.5 | 81.3 |
| | intent-implicit SFT | 85.2 | 78.4 | 93.5 | 90.1 | 78.7 |
| | intent-multiview SFT | 87.0 | 80.2 | 92.2 | 93.3 | 82.5 |

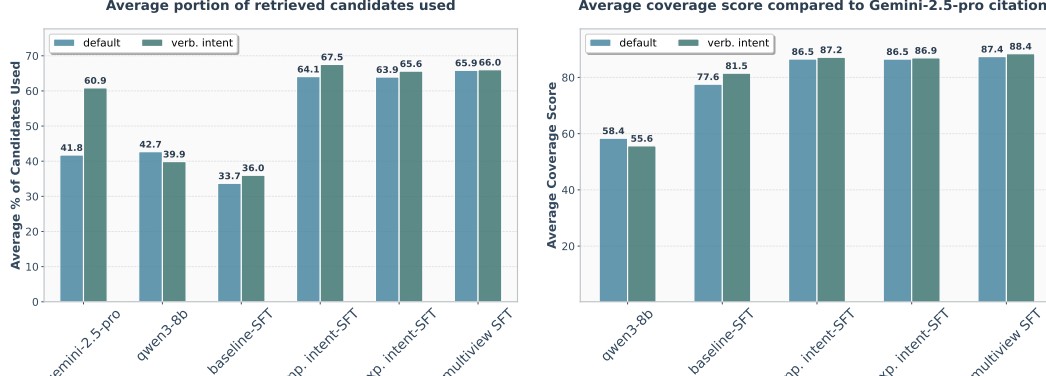

Figure 2: (left) average portion of retrieved candidates used in the generated reports; (right) average citation coverage between small model variants and `gemini-2.5-pro`. All average scores are computed at a query level. *default* and *verb. intent* denotes the different instructions. *verb. intent* denotes the augmentation of intent awareness. The analysis is done on SQA-CS-V2.

**Extended Ablations and Baseline Comparison.** To further validate the performance of verbalized *intents*, we conduct ablations on `gemini-2.5-pro` with different intent categories. Table 5 presents the complementary benefits of both intent categories. On the development set of SQA-CS-V2, citation intents and paragraph intents work orthogonally to result in the best performance. We also compare our inference methods with zero-shot CoT (Wei et al., 2022b; Kojima et al., 2023) prompting and ReAct (Yao et al., 2023). Results show that our intent-aware inference shows better performance when compared to these methods for long-form report generation tasks.

## 4.3 EFFECTIVENESS OF INTENTS IN UNDERSTANDING MODEL BEHAVIOR

**Intent types reveal the model differences in behavior.** We study the distribution of the tag types for citation and paragraph intents in Table 6 , by comparing the model generations with human annotations in the original ACL-ARC (Jurgens et al., 2018) dataset. Overall, the trends align with human annotations, where *Background* and *Uses* emerge as dominant citation intent categories. This suggests that models have learned to capture core citation usage patterns in scholarly writing. There are also a hew notable differences; we see that models significantly underuse *Comparison or Contrast* (around 5%), a category more prevalent in human writing (17%). This gap highlights a limitation in current systems: a tendency to inform or describe rather than synthesize or compare—skills essential for composing useful long-form reports.

Table 5: SQA-CS-V2-dev Performance results with *verbalized intents* and `gemini-2.5-pro`. We **bold** the best row for the Overall metric.

| Method | Variant | Overall | Rubrics | Answer P | Citation P | Citation R |
|---|---|---|---|---|---|---|
| verbalized intent (gemini) | no | 88.1 | 82.6 | 94.1 | 93.2 | 82.4 |
| | all | **89.7** | 82.6 | 94.5 | 95.7 | 86.1 |
| | citation-only | 88.6 | 81.5 | 91.7 | 95.3 | 86.2 |
| | paragraph-only | 89.1 | 82.7 | 92.9 | 95.2 | 85.6 |
| other inference methods | CoT | 81.3 | 71.5 | 94.5 | 83.3 | 76.1 |
| | ReAct | 77.6 | 67.4 | 94.6 | 76.5 | 72.0 |

Table 6: Distribution of the intent types: (left) citation intents and (right) paragraph intents. See Table 1 for the full intent type reference. *others* denotes that the model does not output these pre-defined categories, e.g., just *comparison* for citation intents. We report the human reference from Jurgens et al. (2018) on their ACL-ARC dataset labels, as a reference to general human writing distributions.

| Citation (%) | o3 | gemini | opus-4.1 | *Human ref* |
|---|---|---|---|---|
| Background | 28.2 | 29.6 | 21.1 | 51.9 |
| Motivation | 10.6 | 7.1 | 6.8 | 5.0 |
| Uses | 40.4 | 55.9 | 47.4 | 18.5 |
| Extension | 6.9 | 0.7 | 12.8 | 3.7 |
| Comparison | 4.7 | 4.8 | 3.8 | 17.5 |
| Future | 4.2 | 0.9 | 2.8 | 3.5 |
| (error) | 5.0 | 0.9 | 5.3 | 0.0 |

| Paragraph (%) | o3 | gemini | opus-4.1 |
|---|---|---|---|
| Expos. | 41.5 | 51.5 | 39.9 |
| Def. | 7.0 | 7.1 | 7.3 |
| Argu. | 11.6 | 8.6 | 5.1 |
| Comp.-Contr. | 6.4 | 6.1 | 9.7 |
| Cause-Eff. | 6.1 | 2.6 | 5.4 |
| Prob.-Sol. | 14.5 | 13.4 | 22.8 |
| Narr. | 2.7 | 5.2 | 1.3 |
| Eval. | 9.6 | 5.4 | 8.5 |
| (error) | 0.0 | 0.0 | 0.0 |

We also observe model-specific differences. `gemini-2.5-pro` achieves the best performance but leans heavily on *Uses* (55.9%). It also rarely produces *Extension* or *Future Work* intents, indicating a narrower functional diversity. In contrast, `o3` distributes its citations more evenly, with higher use of *Motivation* and *Future* categories. These differences suggest that intent tagging can help diagnose model tendencies and may guide fine-tuning or evaluation strategies.

**Case Study: Intents help navigate readers in model-generated long-form reports.** To explore the impact of intent-awareness beyond performance on automatic metrics, we conduct a user study to investigate how the presence of intents can shape the users' report-reading experience. Our user study takes a between-subject approach, where some participants read multiple `gemini-2.5-pro`-generated reports from a baseline system, and others read reports generated from our system with intents. To reduce confounds related to the users' prior knowledge and personal interest, participants read reports generated on their own questions. They are instructed to pose/select questions that they (1) genuinely want answered and (2) do not already know the answer to. Details of the system design, interfaces (with screen shots), and the participant pool are introduced in Appendix A.9.

The participants are asked to decide if (1) the displayed information helps them understand whether they want to read this section without opening up the paragraphs; (2) they feel confident that they know what they will learn if they follow the citation and read the cited paper, for each paragraph and highlighted citation, respectively. For each paragraph/highlighted citation, the participants provide a Likert rating on a scale of 1-5 (from Strongly Disagree to Strongly Agree).

In total, we collected labels from 20 participants and 71 reports, with labels for 349 unique paragraphs and 416 unique citations. On average, the participants who read with our systems report $4.47 \pm 0.83$ and $4.46 \pm 0.87$ for paragraph and citation questions, respectively, which suggests that participants generally agree that the intents help them decide whether to read a paragraph in detail or dive into a citation. In contrast, the participants who read with the baseline system report $3.84 \pm 1.05$ and $3.62 \pm 1.18$, which suggests the insufficiency of section titles, first sentences, and supporting snippets alone.

We also qualitatively analyzed participants' optional free-form reflections after they completed the task, which further confirmed that our intent-aware annotations were useful: participants in the experiment condition found the annotations helpful for guiding their reading and their attention span. For example, one participant noted, "Intents are particularly useful when the report includes many hard concepts. Intents help guide the understanding of the relations among the entities". Another

annotator reported that "intent labels (*background, uses, motivation*, etc.)" can "let me quickly judge whether the citation was central to the argument or just providing broader context.", highlighting the usefulness of the schema design. In contrast, participants in the baseline condition found the information overwhelming but still insufficient: "the citation snippet is hard to read and understand the relevance when they are long". These findings highlight the promise of incorporating intent annotations into reading interfaces to support targeted comprehension (Russell et al., 1993; Chang et al., 2023; Lo et al., 2023).

## 5 DISCUSSION

**The Complexity and Hierarchy of Intent.** Our results show that paragraph- and citation-level intents already offer complementary perspectives on the structure of scientific writing. However, we believe these two levels likely only scratch the surface. Human authors often operate with multi-layered, hierarchical intents—where paragraphs build upon one another and citations serve nuanced rhetorical roles (Samraj, 2013; Bhatnagar et al., 2022). For instance, writers may structure paragraphs to contrast ideas or build a multi-step argument, and use citations to critique, anticipate, or contextualize claims. Our schema, though effective, was purely synthetic. We hypothesize that grounding intent schemas in human annotation or behavioral data (e.g., writing process logs, document plans, or outlining strategies) could lead to more sophisticated, accurate modeling of intent hierarchies. Future work could explore tree-structured or graph-based representations of intent to reflect how one paragraph supports, contrasts, or contextualizes another, allowing models to generate globally coherent narratives rather than well-formed but somewhat isolated paragraphs.

**Intent as a Diagnostic and Analysis Layer.** Besides enabling the generation of higher-quality reports, we see that intent awareness provides a new lens for model evaluation and analysis. While existing benchmarks emphasize factuality and citation correctness, they often miss why and how content is organized. In contrast, our intent-centric analysis already helps highlight the distributional differences between human- and model-written texts in Sec. 4.3 (e.g., human writing include significantly more comparisons). This suggests that intents can help inform the design of new benchmarks or scoring rubrics that reward desirable patterns of argumentation, such as balanced comparisons, causality chains, or synthesis of conflicting evidence, so as to distinguish models that have strong capability to synthesize complex information beyond factual lists. Intent scaffolding may also support self-critique or refinement loops, where models justify and revise their own structure.

**Generalization Across Domains.** Our study focused on scientific domains, where writing tends to follow conventional structures. However, in other disciplines such as policy, law, or the humanities, the nature of intent types may vary considerably (Harrington et al., 2019; Lafia et al., 2023). Citations might serve rhetorical, historical, or ethical functions that our current schema does not capture. To generalize, future work is needed for understanding how intent distributions vary by domain, whether schemas need to be domain-adaptive, and how models might learn new intent categories from domain-specific corpora.

## 6 CONCLUSION

Drawing inspiration from the human writing process, we develop an intent-aware writing framework that helps language models produce better quality reports for scientific deep research tasks. Our strategies of incorporating intent awareness, during both inference and training, lead to improved model performance across several challenging benchmarks. We further showed that training data generated with intent awareness can be used for distillation, enabling the smaller base models to match state-of-the-art larger model performance. We demonstrate potential utility beyond automatic metrics: a case study with researchers suggests that our intents can potentially aid reading comprehension and efficiency. More broadly, our results provide preliminary yet encouraging evidence that incorporating elements of human writing processes—especially those missing from data used to train language models—can enhance their text generation capabilities. We open-source our code and model checkpoints to encourage further research in this area.

ACKNOWLEDGMENTS

The authors thank Hongming Zhang, Sihao Chen, Tong Chen, Runlong Zhou, Ryan Liu, Shannon Shen, Boyuan Zheng, Ken Liu, Xiang Li, Yiming Zhang, as well as other AI2 interns (including but not limited to Yue Yang, Hita Kambhamettu, Yapei Chang, Federica Bologna, Amanda Bertsch, Michael Noukhovitch, Nishant Balepur, Peiling Jiang, Alexiss Ross, Ruochen Li, and Anej Svete) and UW students (including but not limited to Scott Geng, Rui Xin, Rulin Shao, Zhiyuan Zhang, Hamish Ivison, and Oscar Yin), for their insights into design and evaluation choices. The authors sincerely appreciate the constructive discussions with colleagues from CMU WInE Lab. The authors also thank the anonymous reviewers and area chair for helpful discussions and comments. At CMU, Xinran Zhao is supported by the ONR Award N000142312840 and the Amazon AI PhD Fellowship.

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

## A   APPENDIX

### A.1   IMPLEMENTATION DETAILS.

For Gemini, Claude, and GPT models, we use the official API service. For other open-sourced models, we use our locally served model on nodes with 8 Nvidia H100 (80G) GPUs with CUDA 12 installed, with an inference structure built upon SGLang (Zheng et al., 2024). If applicable, we set the max output token to be 22,000, the temperature to be 1.0. If not further specified, we use the original hyperparameters and settings when evaluating the tasks. Following the original LLM-as-a-judge choice. We use `gemini-2.5-flash` for AstaBench-SQA-CS-V2; `gpt-4o` for DeepScholar Bench, and `gpt-4.1-mini` for ResearchQA.

For fine-tuning, we use $5e - 6$ learning rate, 80 training epochs, and 4 gradient accumulation steps for all base models and variants, unless further specified. We generated training data from Gemini-2.5-pro with our inference pipeline; this model was within 2 points of the best-performing model (Claude-4-Opus) on all datasets when we generated answers while eliciting intents at inference time, while being an order of magnitude cheaper. We use the inference prompt in Appendix A.6 to collect the training data.

### A.2   THE USE OF LARGE LANGUAGE MODELS (LLMS).

In this work, LLMs are used for correcting grammatical errors in writing and coding. We do not use LLMs to write papers or construct the logic of the whole code base.

### A.3   FULL PERFORMANCE FOR SFT VARIANTS

In the main paper, we mainly discuss the performance with intent-aware prompts. Here we further compare the performance difference across different fine-tuned small models in Table 7. We can observe consistent findings in the main content: in most cases, augmenting the models with intent awareness at test time helps improve model performance, especially for fine-tuned models.

### A.4   PRE-PLANNING WITH INTENTS

In our proposed method, the intents are generated in-line with the rest of the text. We also tested a variant where we do pre-planning by generating potential citation intents and a relevance score for each retrieved paper before generating the answer. The retrieved papers are reordered according to the generated citation scores and the answer is generated conditioned on the reordered retrieved documents and their corresponding intents. The results from this setting are shown in Table 8.

### A.5   ETHICAL STATEMENTS

We foresee no ethical concerns or potential risks in our work. All datasets are open-sourced, as shown in Section 4.1. The LLMs we applied in the experiments are also publicly available. Given our context (long-form report generation with queries verified by humans), the outputs of LLMs are unlikely to contain harmful and dangerous information. The experiments in our paper are mainly on English.

Table 7: SQA-CS-V2 Performance Results Across Base Models and Variants

| Base Model | Variant | Overall | Rubrics | Answer P | Citation P | Citation R |
|---|---|---|---|---|---|---|
| gemini-2.5-pro(ref) | - | 88.1 | 82.6 | 94.1 | 93.2 | 82.4 |
| | no training | 80.7 | 82.1 | 90.4 | 83.2 | 66.9 |
| | -verb. intent | 80.9 | 80.2 | 92.6 | 81.3 | 69.8 |
| qwen3-8b | SFT | 83.2 | 78.7 | 94.3 | 85.8 | 73.9 |
| | -verb. intent | 84.6 | 79.0 | 94.6 | 87.6 | 76.9 |
| | intent-explicit SFT | 86.7 | 79.4 | 91.7 | 92.3 | 83.6 |
| | -verb. intent | 88.0 | 80.5 | 93.0 | 93.6 | 85.0 |
| | intent-implicit SFT | 86.7 | 77.9 | 91.1 | 93.7 | 83.9 |
| | -verb. intent | 87.1 | 78.9 | 94.0 | 92.5 | 82.9 |
| | intent-multiview SFT | 87.9 | 79.2 | 93.6 | 94.1 | 84.7 |
| | -verb. intent | **88.6** | 81.4 | 94.7 | 93.7 | 84.7 |
| | no training | 66.4 | 64.6 | 77.5 | 67.2 | 56.1 |
| | -verb. intent | 64.7 | 59.5 | 86.1 | 63.2 | 49.8 |
| llama-3.1-8B | SFT | 84.4 | 78.1 | 92.3 | 89.8 | 77.4 |
| | -verb. intent | 85.5 | 78.4 | 93.8 | 89.9 | 79.9 |
| | intent-explicit SFT | 87.8 | 80.1 | 93.1 | 93.4 | 84.8 |
| | -verb. intent | 85.8 | 77.6 | 93.1 | 90.5 | 82.2 |
| | intent-implicit SFT | 87.2 | 79.0 | 92.3 | 93.2 | 84.3 |
| | -verb. intent | 87.8 | 77.9 | 93.3 | 94.0 | 85.9 |
| | intent-multiview SFT | 87.5 | 77.3 | 93.9 | 93.8 | 85.0 |
| | -verb. intent | **89.2** | 79.5 | 95.1 | 95.4 | 86.7 |
| | no training | 80.9 | 78.0 | 94.6 | 82.8 | 68.1 |
| | -verb. intent | 80.2 | 78.6 | 94.7 | 80.7 | 67.0 |
| qwen3-4b | SFT | 83.4 | 80.1 | 92.4 | 86.2 | 74.8 |
| | -verb. intent | 86.7 | 77.3 | 93.2 | 92.5 | 83.6 |
| | intent-explicit SFT | 86.3 | 80.4 | 91.3 | 92.2 | 81.5 |
| | -verb. intent | 87.5 | 80.1 | 97.0 | 91.5 | 81.3 |
| | intent-implicit SFT | 83.7 | 77.0 | 92.8 | 88.0 | 77.0 |
| | -verb. intent | 85.2 | 78.4 | 93.5 | 90.1 | 78.7 |
| | intent-multiview SFT | **87.9** | 79.0 | 93.7 | 93.7 | 85.2 |
| | -verb. intent | 87.0 | 80.2 | 92.2 | 93.3 | 82.5 |

Table 8: SQA-CS-V2 Performance Results with Pre-planning

| Base Model | Variant | Overall | Rubrics | Answer P | Citation P | Citation R |
|---|---|---|---|---|---|---|
| | default | 85.1 | 91.4 | 96.5 | 89.4 | 63.4 |
| o3 | + pre-planning | 86.5 | 90.5 | 95.1 | 91.7 | 68.8 |
| | + intents | 86.2 | 89.6 | 95.1 | 90.5 | 69.8 |
| | default | 88.1 | 82.6 | 94.1 | 93.2 | 82.4 |
| gemini-2.5-pro | + pre-planning | 88.4 | 81.3 | 93.7 | 93.4 | 85.3 |
| | + intents | 90.7 | 80.6 | 94.0 | 97.2 | 90.9 |
| | Default | 85.4 | 84.3 | 87.9 | 89.6 | 79.6 |
| claude-opus-4.1 | + pre-planning | 89.4 | 85.3 | 93.9 | 93.8 | 84.7 |
| | + intents | 90.0 | 85.2 | 93.1 | 95.7 | 86.2 |

### A.6 PROMPTS USED

We present the example prompt we used for *verbalized intents* below. During inference, retrieved information will be provided to the model by replacing {section_references}. Each snippet in {section_references} will be in the format of "[Citation X] Snippet", and the model is instructed to cite the relevant references.

---

A user issued a query and a set of research papers were provided with salient content. The user query was: query

I will provide you with a list of chosen quotes from these papers that may be relevant to the user query. It's important to note that the quotes may *not* be relevant. Carefully consider this before adding them to the answer.

Your job is to help me write a multi-section answer to the query and cite the provided relevant quoted references. Cite all of the *relevant* quoted references. Exclude all of the irrelevant quoted references from your answer.

Here are the relevant reference quotes to cite: section_references {section_references}

Citation Instructions:

- Each reference quote (section) is a key value pair, where the key is in the form "[Citation 'int']". You should cite 'int' when referring to any of these sections as evidence.

- Please write the answer, making sure to cite the relevant references inline using the corresponding reference key in the format: [CitationNumber]. You may use more than one reference key in a row if it's appropriate but no more than five references in a row. In general, use all of the references that support your written text, but cite no more than five references in a row. Having more than five references or citations at a time overwhelms the user, so only include up to the five most relevant.

- For each reference you cited in the section content, be sure to carefully consider the intent of the citation. Your citation intent must be expressed in the format of: your description <bcit> [citation intent Type]: your rationale <ecit> [Citation 'int']... The Type ([citation intent Type]) should be a single, capitalized word from the list below, and the rationale should be a brief explanation of why the citation is used in this context given the type. Only use one type per citation and add your own type if none of the types fit.

Here is a list of the potential citation intent types:
(1) CIT-BACKGROUND: the citation provides relevant information for this domain;

(2) CIT-MOTIVATION: the citation illustrates need for data, goals, methods, etc.;

(3) CIT-USES: the sentence uses data, methods, etc. from the citation;

(4) CIT-EXTENSION: the sentence extends the referenced work's data, methods, etc. of the citation;

(5) CIT-COMPARISON OR CONTRAST: the sentence expresses similarity/differences to the referenced work of the citation;

(6) CIT-FUTURE: the citation identifies the referenced work as a potential avenue for future work.

- The rationale wrapped in <bcit><ecit> should be a brief and contextual explanation of what text in the quote triggers the citation.

---

- **Do not** repeat the information and text that is already in the citing sentence.

- Your rationale should use or summarize the relevant part of the reference quote you are citing and connect it to the citing sentence.

- You should write **different** citation intents for each citations even if they are in the same sentence or have the same type.

- Your citation intent should potentially help the reader understand why you are citing the reference quote and what they could potentially learn from further reading the cited paper.

- Along with the quote, if any of its accompanying inline citations are relevant to or mentioned in the claim you are writing, you should cite the reference of the section (i.e. the integer in [Citation 'int'])

- if you are using multiple citations, you should write separate citation intents for each of the citations, although you can have the same type for multiple citations.

- You can add something from your own knowledge. This should only be done if you are sure about its truth and if there is not enough information in the references to answer the user's question. Cite the text from your knowledge as [LLM MEMORY | 2025]. The citation should follow AFTER the text. Don't cite LLM Memory with another evidence source.

- Note that all citations that support what you write must come after the text you write. That's how humans read in-line cited text. First text, then the citation intent tag, then the citation.

---

Writing instructions: Guidance for organizing content:

- Write a well-organized narrative that flows logically, with clear structure and coherence between ideas.

- The answer should be written in sections that break down the user query for a scientific audience.

- Each section should discuss a **dimension or theme** that is related to the query.

- Most sections will correspond to a cluster of related quotes that comprise of **similar claims, shared concepts, or overlapping evidence**. If multiple quotes from different citations support the same idea or theme, they should be grouped and cited together in one section.

- Be sure to carefully consider your intents to write each paragraph in the section.

Each section should have the following characteristics:
- Before the section write a 2 sentence "TLDR;" of the section. No citations here. Precede with the text "TLDR;"

- The first section should almost always be "Background" or "Introduction" to provide the user the key basics needed to understand the rest of the answer.

- Every section can contain multiple paragraphs and should correspond to a theme or dimension.

- Use multiple paragraph to organize the content within each section.

- Each paragraph should focus on a central high-level idea and should correspond to a cluster of similar citations.

- Be sure to carefully consider your intents to write each paragraph. Before each paragraph within the text field, you must insert a paragraph intent tag in the format: <bpit>[paragraph intent Type]: Rationale... <epit>.

The [paragraph intent Type] should be a single, capitalized word from the list provided below extracted from research about discourse mode. The Rationale should be a brief explanation of why the paragraph fits the chosen type, based on its content and function within the report.

Here is a list of potential paragraph intent [paragraph intent Type]s and their descriptions:

(1) PIT-Exposition: This paragraph's main function is to explain, clarify, or provide background information on a topic (e.g., introducing a concept, summarizing prior work).

(2) PIT-Definition: This paragraph's primary purpose is to define a key term, concept, or theory, often providing necessary boundaries for its use in the report.

(3) PIT-Argumentation: This paragraph presents a specific claim or thesis and supports it with evidence, logic, or reasoning to persuade the reader.

(4) PIT-Compare-Contrast: This paragraph's structure is organized around highlighting the similarities and/or differences between two or more subjects, theories, or findings.

(5) PIT-Cause-Effect: This paragraph focuses on explaining the causal relationship between events or phenomena, detailing why something happened or what its results were.

(6) PIT-Problem-Solution: This paragraph identifies a specific problem, gap, or challenge and then proposes or describes a potential solution or response.

(7) PIT-Evaluation: This paragraph assesses the strengths, weaknesses, validity, or significance of a study, theory, or piece of evidence according to a set of criteria.

(8) PIT-Narration: This paragraph recounts a sequence of events, such as the historical development of a field, the chronology of a case study, or the steps in a process.

For example, you can write:
<bpit>[PIT-Exposition] This paragraph provides background context by introducing Convolutional Neural Networks (CNNs) and stating their established success in image classification, setting the stage for the subsequent discussion. <epit> Convolutional neural networks (CNNs) have achieved state-of-the-art results in image classification <bcit>[CIT-BACKGROUND]: these citations provides foundational context linking CNN to major image classification tasks <ecit> [1] [2]. They have become a foundational tool...

- Use direct and simple language everywhere, like "use" and "can". Avoid using more complex words if simple ones will do. Use the citation count to decide what is "notable" or "important". If the citation count is 100 or more, you are allowed to use value judgments like "notable."

- Some references are older. Something that claims to be "state of the art" but is from 2020 may not be any more. Please avoid making such claims that may no longer be true.

- The answer should directly respond to the user query. Every paragraph should be directly relevant to the user query. If the user asked about "Visual RAG", don't write a paragraph about just RAG unless it's in the one background section.

Format Instructions

When references present conflicting findings or contradictory claims:

- Explicitly acknowledge the disagreement rather than ignoring it. Use phrases like "While X et al. found..., Y et al. reported contrasting results..."

- Present both/all perspectives with their respective citations

- If possible, identify potential reasons for the discrepancy (e.g., different methodologies, sample sizes, time periods, or contexts)

- Use citation counts as one indicator of relative weight, but do not dismiss lower-cited work solely on this basis

- If one claim has substantially more supporting evidence across multiple papers, you may note this: "The majority of studies support..." while still acknowledging the minority view

- Avoid taking sides unless the evidence overwhelmingly supports one position

- If the conflict is central to answering the user's query, consider dedicating a section to "Conflicting Findings" or "Ongoing Debates"

Start the section with a 'SECTION;' marker followed by its section name and then a newline and then the text "TLDR;", the actual TLDR, and then write the summary.

Write the section content using markdown format.

Rules for section formatting:

- For each section, decide if it should be a bullet-point list or a synthesis paragraph.

- Bullet-point lists are right when the user wants a list or table of items.

- Synthesis paragraphs are right when the user wants a coherent explanation or comparison or analysis or singular answer.

- Use section names to judge what section format would be best. Lists and syntheses paragraphs are the only allowed formats.

- Remember to include both citation intents (<bcit> and <ecit>) and paragraph intents (<bpit> and <epit>) in your answer.

Table 9: Extended performance comparison on Deepscholar Bench. **Bold** indicates the best-performing row for overall metrics.

| Method | DeepScholar Bench (DSB) | | | | |
|---|---|---|---|---|---|
| | Overall | Nug. Cov. | Org. | Cite-P | Claim Cov |
| o3 | 46.8 | 47.0 | 61.1 | 39.1 | 40.2 |
| + intent | 43.2 | 49.1 | 64.1 | 27.2 | 34.3 |
| + intent (paragraph-only) | **49.3** | 48.0 | 66.0 | 39.1 | 44.3 |
| qwen3-8b | 56.0 | 46.0 | 59.0 | 57.0 | 62.0 |
| intent-explicit | 59.5 | 48.2 | 68.0 | 61.2 | 63.1 |
| intent-implicit | 60.3 | 45.1 | 68.0 | 63.1 | 65.0 |
| intent-multiview | 57.5 | 45.1 | 60.0 | 62.3 | 63.1 |

Table 10: SQA-CS-V2-dev Performance results with *verbalized intents* and gemini-2.5-pro. We compare variants of intent schema design. *free* denotes the use of model improvised types. *current* denotes the use of our schema. *mix* denotes the use of most frequent types in our schema and let the model has freedom on adding their own. We **bold** the best row for the Overall metric.

| Method | Variant | Overall | Rubrics | Answer P | Citation P | Citation R |
|---|---|---|---|---|---|---|
| | free | 89.3 | 82.4 | 92.0 | 96.1 | 86.7 |
| verbalized intent (gemini) | current | 89.7 | 82.6 | 94.5 | 95.7 | 86.1 |
| | mix | **91.6** | 83.1 | 95.0 | 97.3 | 91.0 |

## A.7 Extended DeepScholar Bench Results

In the main paper, we found that, while o3 has much worse citation behavior than other frontier models on DeepScholar Bench, our intent-aware inference will degrade the citation quality. To further validate the performance of our method, we report the performance on DeepScholar Bench with our paragraph-intent-only inference and SFT variants in Table 9. The variants are with the same setting as in Table 4 and Table 5 in the main paper, without further training.

Results show that our models generalize to DeepScholar Bench: the best performing variant (intent-implicit) achieves better performance than the best performing large model (opus-4) in our Table 3. On the citation metrics, our SFT variants generally show better overall scores compared to o3 as well.

## A.8 Further Ablation on Intent Schema Design

We further design a variant of our intent-aware inference with a more dynamic schema: we only keep the top-3 most used types for citation and paragraph intents from Table 6, respectively, in the instruction, and ask the model to improvise if necessary. We denote this variant as intent (mix) and compare with the current version, i.e., intent (current), and an ablated variant, intent (free), where the model outputs their own types.

We observe that keeping the most frequent types in our schema + extra freedom (i.e., intent (mix)) would lead to the best performance. We will update these experiments in the appendix as an alternative design. On the other hand, given that most of our design is kept the same (type + rationale), inference without a pre-set type leads to similar performance as with model-improvised types.

However, beyond the performance, the types used will be inconsistent across questions for intent (mix) and intent (free type) as a trade-off of the freedom, e.g., [Example] vs. [Exemplify] vs. [Instance]. The intent (current) variant, where we extract a unified schema for all questions extracted from literature, has its value in providing consistent types for analysis and readability.

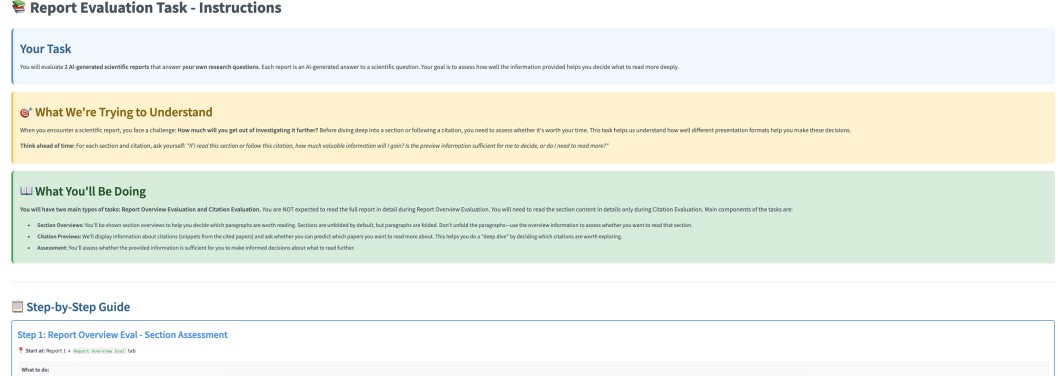

Figure 3: A screenshot of the instructions to the users. Besides the tasks shown in the figure, the users will also be provided with a step-by-step guide on the annotation tasks and the key points to remember. The instructions can be revisited during the annotation task by the users by clicking a "Click to Expand Instruction" button.

## A.9 USER STUDY DETAILS

The participants are first introduced to the instructions, background of the report generation tasks, and the schemes for our intents. Then, they will start an interactive session with random intent text and the corresponding context to answer questions one by one. They are also allowed to provide optional qualitative feedback.

Baseline systems present (in GUI): (1) for paragraphs, the section titles and first sentences, with the full content folded; (2) for citations, relevant snippets from the papers cited are inline in tooltips that appear when hovering over the citation. Our system presents automatically generated PITs (before each paragraph) and CITs (before the snippet) in our experimental condition. The participants are asked to decide if (1) the displayed information helps them understand whether they want to read this section without opening up the paragraphs; (2) they feel confident that they know what they will learn if they dive into the citation, for each paragraph and highlighted citation, respectively. For each paragraph/highlighted citation, the participants provide a Likert rating on a scale of 1-5 (from Strongly Disagree to Strongly Agree). The screenshots of the systems are shown in Figure 3 (general instructions), Figure 4 (PIT questions for the baseline system), Figure 5 (CIT questions for the baseline system), Figure 6 (PIT questions for our intent-aware system), and Figure 7 (CIT questions for our intent-aware system)).

**Participant pool.** We are recruiting participants with a master's or PhD in computer science to obtain more diverse and representative expertise. In this round, we recruited from two sources: (1) Personal advertising: we recruited 8 participants from 7 affiliations; (2) Prolific [3]: we recruited 12 participants. Each annotator receives compensation of 30 USD per hour. All participants are new to the task. We rule out annotators whose average score is outside 2 standard deviations of the mean of all other annotators or who spent significantly less time than others, e.g., less than 2 minutes.

**User-centered task design.** To reduce confounds related to the users' prior knowledge and personal interest, participants read reports generated on their own questions. They are instructed to pose/select questions that they (1) genuinely want answered and (2) do not already know the answer to.

Besides the results reported in main paper, there is also high consistency in the findings with participants hired from different sources: (1) From personal advertising, 8 participants reading with our system report 4.26 and 4.19 for paragraph and citation questions, while the scores are 3.77 and 3.29 for the baseline system; (2) From Prolific, 12 participants report 4.55 and 4.60 for paragraph and citation questions reading reports with our system, while 3.94 and 4.06 reading with the baseline system. Consistent annotation results from different demographics strengthen the claim in our original case study that the intent annotations in reading interfaces help support targeted comprehension.

---

[3] https://www.prolific.com/

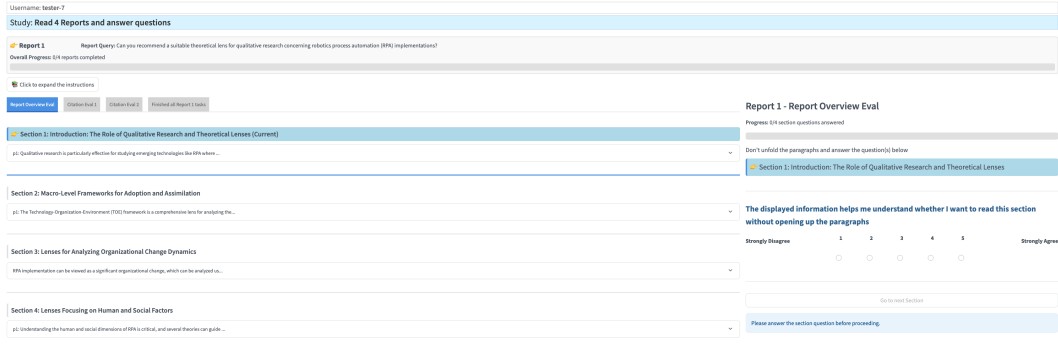

Figure 4: A screenshot of the PIT questions for the baseline system. Users are shown with the section titles and paragraph first sentences to answer the questions.

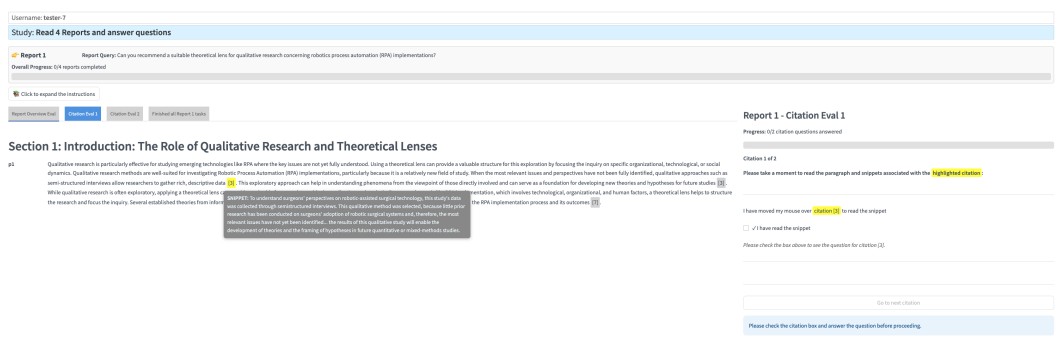

Figure 5: A screenshot of the CIT question for the baseline system. Users are shown with a specific paragraph with one highlighted citation to answer the question. The snippet will show as the users move their mouse over it.

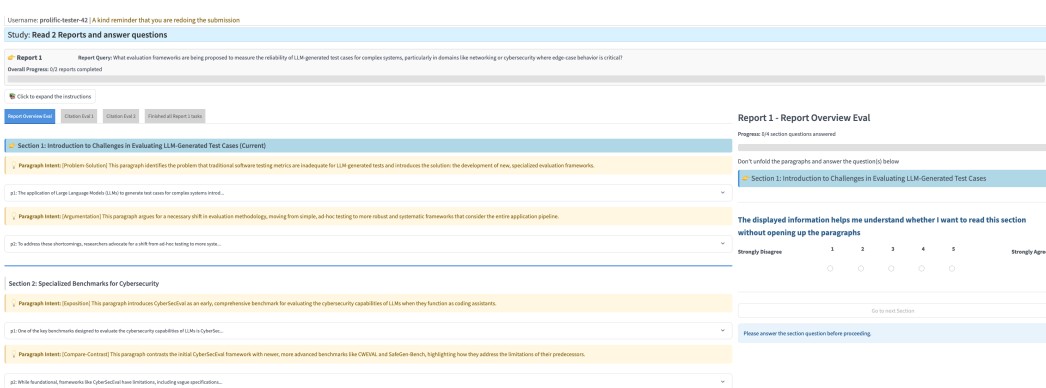

Figure 6: A screenshot of the PIT questions for our intent-aware reading system. Users are shown with the section titles, the paragraph-level intents, and paragraph first sentences to answer the questions.

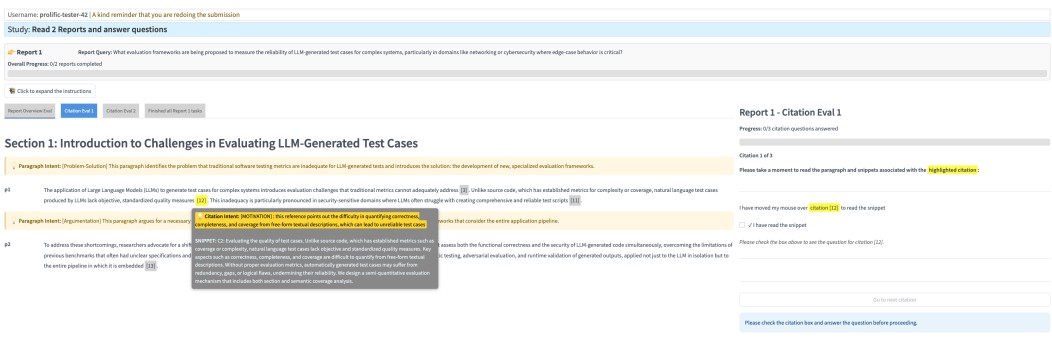

Figure 7: A screenshot of the CIT question for the intent-aware reading system. Users are shown with a specific paragraph with one highlighted citation to answer the question. The potential citation intent and the snippet will show as the users move their mouse over it.

