# OpenReview forum: "Improving Attributed Long-form Question Answering with Intent Awareness"
_ICLR.cc/2026/Conference — ICLR 2026 Poster_

### Official Review · Reviewer_yQe7 · 2025-10-28

**Soundness:** 3
**Presentation:** 3
**Contribution:** 3
**Rating:** 6
**Confidence:** 4

**Summary:**

The paper introduces an _intent-aware writing framework_ for attributed long-form question answering.  It augments both inference and training with explicit paragraph- and citation-level intent tags that describe the purpose of text segments (e.g., "background," "motivation," "cause–effect").   Experiments on three benchmarks—SQA-CS-V2, DeepScholar, and ResearchQA—show consistent gains  (+2.9 for large LMs, +12.3 for small ones).

**Strengths:**

1. Clear motivation linking human writing processes to LLM report generation, supported by consistent empirical evidence.
2. Simple but effective method with broad applicability—plug-and-play intent tagging improves attribution, readability, and transparency.

**Weaknesses:**

1. The conceptual novelty is limited — effectively a structured form of rationale or CoT distillation.
2. Intent schema is static and handcrafted, not learned or generalized across domains.
3. Lacks comparison to stronger baselines such as _CoT + Answer_ or _rationale distillation_.

**Questions:**

1. How does intent-aware SFT compare to _CoT + Answer_ distillation under equal data and compute budgets?
2. If the model could _self-summarize intents_ from non-intent data (i.e., induce tags automatically) before SFT, would this remain effective—transforming the contribution from distillation to curation innovation?
3. Can the approach generalize to less-structured domains such as policy or humanities writing?

---

> ### Author Response · Authors · 2025-11-24
> **Thank you for the review! - Part 1**
>
> We are grateful for your constructive feedback and positive comments, especially on the simplicity and effectiveness of our proposed method. We address your questions below:
>
> **W1, W3, and Q1: Comparison with CoT**
>
> Conceptually, our work is motivated by CoT distillation work such as STaR and ToW, as discussed in related work (Line 122-128).
>
> However, while previous work mostly focuses on short-form QA and math reasoning tasks, we aim to tackle long-form QA, specifically deep research-style report generation. This task requires dealing with long contexts (total input + output length is typically 17,000-22,000 tokens) and ensuring faithful attribution of claims. Hence, we propose to use specific types of rationales targeting these two issues: (i) paragraph intents and rationales for better organization of long output, and (ii) citation intents and rationales for better attribution. Additionally, instead of generating a single rationale/chain of thought before the report, our method allows targeted in-line thought/rationale generation before every paragraph or citation.
>
> Following your suggestions, we evaluated (1) zero-shot CoT [1][2] and (2) ReAct [3] baselines on the AstaBench-SQA-CS-V2 dataset, using Gemini 2.5 Pro as the backing LLM; results below:
>
> | Method | Overall | Rubrics | Answer Precision| Citation Precision | Citation Recall |
> | :-------------------: | :-: | :-: | :-: | :-: | :-: |
> | CoT | 81.3 | 71.5 | 94.5 | 83.3 | 76.1 |
> | ReAct | 77.6 | 67.4 | 94.6 | 76.5 | 72.0 |
> | Intent (ours) | 89.7 | 82.6 | 94.5 | 95.7 | 86.1 |
>
> We see that our method improves significantly over CoT/ReAct baselines, indicating that our design of rationale types targeting issues inherent to long-form QA is highly effective. With the ReAct baseline in particular, we often see the model start to generate irrelevant markdown comments and tables that further degrade performance. Given the large performance gap between these methods in the inference-only setting, we did not conduct any distillation experiments using data generated from CoT or ReAct-based teachers.
>
> **W2: Intent Schema Design**
>
> To assess the performance of our method under a more flexible schema design, we test the following intent-aware inference variant: we only retain the top-3 most used citation and paragraph intent types in the instruction, and ask the model to improvise additional types if necessary. The table below compares this variant, denoted as Intent (mix), with our current method (Intent (current)), and another ablated variant (Intent (free type)), in which the model is allowed to freely output intent types without any restrictions, on AstaBench-SQA-CS-V2, using Gemini 2.5 Pro as the backing LLM:
>
> | Method | Overall | Rubrics | Answer Precision| Citation Precition | Citation Recall |
> | :-------------------: | :-: | :-: | :-: | :-: | :-: |
> | Intent (free type) | 89.3 | 82.4 | 92.0 | 96.1 | 86.7 |
> | Intent (current) | 89.7 | 82.6 | 94.5 | 95.7 | 86.1 |
> | Intent (mix) | 91.6 | 83.1 | 95.0 | 97.3 | 91.0 |
>
> We observe that retaining the top-3 most frequent types from our schema and allowing flexibility for remaining types (i.e., Intent (mix)) performs best, while allowing completely free-form intents leads to similar performance as our current method. This indicates that our method can work with more flexible/underspecified schemas, which would help it generalize to other domains easily. We have added these experiments to Appendix A.7.
>
> However, allowing flexible intent types has some drawbacks. For instance, if these intents are surfaced to users perusing long-form reports to aid with readability, allowing flexible intent types can lead to inconsistencies in their use across questions (e.g., using [Example], [Exemplify], [Instance] interchangeably for the same intent). Our current variant, which uses a unified schema across all questions, provides consistent types, balancing performance with downstream benefits like better readability and ease of analysis.
>
> **Q2:  Self-summarizing Intents**
> Could you clarify this a little? Here is our current understanding: we take answers that were not generated in an intent-aware manner, have an LLM annotate them post-hoc with intent tags and rationales, and then run SFT on this data. If this is different from what you had in mind, please let us know, and we will be happy to run the requested ablation, if feasible, during the rebuttal time frame!

---

> > ### Author Response · Authors · 2025-11-24
> > **Thank you for the review! - Part 2**
> >
> > **Q3: Generalization to other domains of writing**
> >
> > We primarily focus on scientific writing in this work, but it would be exciting to see future work test our technique in other, less-structured domains such as policy or humanities writing, as discussed in the original Section 5. Our intent schema could be re-designed for other domains, or a more flexible variant of our intent-aware method (allowing free-form intent types) could be used; we should above that both methods are equally performant.
> >
> > Thank you for your time and effort in reviewing our paper. We have incorporated the experiments you suggested and believe they have strengthened our work. We hope you increase your rating in light of these revisions. Please let us know if you have any further questions.
> >
> >
> > References
> >
> > [1] Large Language Models are Zero-Shot Reasoners (https://arxiv.org/abs/2205.11916)
> >
> > [2] Chain-of-Thought Prompting Elicits Reasoning in Large Language Models (https://arxiv.org/abs/2201.11903)
> >
> > [3] ReAct: Synergizing Reasoning and Acting in Language Models (https://arxiv.org/abs/2210.03629)

---

### Official Review · Reviewer_yUc4 · 2025-10-31

**Soundness:** 3
**Presentation:** 4
**Contribution:** 2
**Rating:** 6
**Confidence:** 4

**Summary:**

The paper proposes an intent-aware report generation framework that can be used for both prompting and fine-tuning. The framework includes paragraph intent and citation intent (sentence-level), explaining the intent of including the citation or paragraph. In prompting, the model first generates the intent, and then generates the actual paragraph or citation. In fine-tuning, the student model is trained on data generated with a larger teacher model. Results show this improves citation precision / recall. Analysis shows that LMs capture the general idea of how people use intents, but distribution could vary among LMs. Small-scale user study shows that this method improves readability of the generated report.

**Strengths:**

1. The paper is clearly written and easy to follow. I can read the paper without constantly referring to details in the Appendix.
2. The proposed intent-aware prompting / fine-tuning methods improve Citation Precision and Recall.
3. Clear ablation study showing the importance of each component.
4. Interesting analysis showing how LMs generate reports with different proportions of intents. There is also potential for this to be an analysis / evaluation tool in the future as mentioned by the authors. I almost feel like this is the most exciting part of the paper.
5. The human study including three PhD students reading actual generated reports is also interesting, showing the potential of this framework for more readable reports. Although I wish larger scale experiments could be done in this direction (improving readability).

**Weaknesses:**

1. Baselines are lacking. For example, I think at least a simple Chain-of-Thought (CoT) prompting baseline should be included. For fine-tuning, training on CoT generated by teacher models should also be included. There should be some additional baselines, like STaR (Zelikman et al., 2022) for example.
2. The improvements mainly come from Citation Precision / Recall. For SQA-CS-V2 (Table 2), only Claude improves on Rubrics with intent-aware prompting. However in Table 3, Claude does not improve on Nugget Coverage. The same can be said for fine-tuning results (Table 4). It is also unclear why this is the case, given the paragraph-only variant (Table 5) also improves about 3 points in Citation Precision and Recall, but only 0.1 in Rubrics and -1.2 in Answer Precision.
3. There is no significance test of any sort. Given some of the performance gains are pretty small, I think statistical significance is needed for judging the effectiveness of the method.
4. Following 2., I think the claims that “performance” or "capability" in generating better reports seem to be overstatements, given the gain mostly come from better citation usage. I don’t know exactly how the “Overall” score is computed (not mentioned in the paper), but most of the gains in overall score seem to come from citation P/R. I think it would be better if the authors are more straightforward about this in the abstract and introduction. Combining my previous points, I almost feel like the advantage of using this framework is not in the performance, but in better citation usage and potentially better readability as shown in the case study.
4. Minor Points:
a. Some inconsistencies in experiments. Why not include o3 in Table 3? Why not include Deepscholar Bench and ResearchQA in Table 4? These should be explained.
b. In the paragraph “intent awareness improves citation usage”, from L370 to L372, the claim that intent-aware prompting “significantly improves the portion of retrieval candidates used” is not accurate. The only gain that I think is significant is on gemini-2.5-pro. Also, the claim of “without precision loss” is also not supported for the Qwen models.

Reference:
Zelikman, Eric, et al. "Star: Bootstrapping reasoning with reasoning." Advances in Neural Information Processing Systems 35 (2022): 15476-15488.

**Questions:**

1. How is "Overall" score computed for all the datasets?
2. Why not include o3 in Table 3? Why not include Deepscholar Bench and ResearchQA in Table 4? (Same as mentioned in 5. in Weakness)
3. Is more citation always better? For example, if there are multiple works showing RAG helps factoid QA, is citing 10 of them better than just citing 5 of them? I wonder how much the claim that "increased citation usage denotes that the model can cover more diverse points"  (L372-373) is true.

---

> ### Author Response · Authors · 2025-11-24
> **Thank you for the review! - Part 1**
>
> Thank you for your insightful and detailed feedback. We address your questions below.
>
> **W1: CoT Baseline**
>
> Thank you for the suggestion. We agree that a CoT prompting baseline is important to understand whether our proposed intent structure is valuable. Specifically, we add (1) zero-shot CoT [1][2] and (2) ReAct [3] baselines with our experimental setup on SQACS-V2 with Gemini 2.5 Pro:
>
> | Method | Overall | Rubrics | Answer Precision| Citation Precision | Citation Recall |
> | :-------------------: | :-: | :-: | :-: | :-: | :-: |
> | CoT | 81.3 | 71.5 | 94.5 | 83.3 | 76.1 |
> | ReAct | 77.6 | 67.4 | 94.6 | 76.5 | 72.0 |
> | Intent (ours) | 89.7 | 82.6 | 94.5 | 95.7 | 86.1 |
>
> As shown in the table above comparing the inference techniques, our method shows significant performance improvement over the baselines. Given the large performance gap from the pilot analysis on different inference performances, we did not proceed with the (STaR-like) distillation on CoT nor ReAct since our SFT variants (e.g., intent-multiview qwen3-8B in Table 3, with 88.6 overall) typically show on par performance with Gemini 2.5 pro with intent-aware inference. We have added these baselines and the corresponding results to Section 4.2 of the revised paper.
>
> **W2:  Improvements of different sub-metrics**
>
> In our experiments with state-of-the-art LLMs, we see that metrics like rubric score, nugget coverage, and answer precision, which do not consider citation quality, do not move much in Tables 2 and 3 because such LLMs are already highly capable of extracting key facts from retrieved information and ensuring that the presented information is topically relevant to the query. However, these models are not very good at attributing the claims made in the answer. Therefore, we expect the biggest impact of using citation intents to be improving models’ attribution capabilities, and we are seeing significant improvement in citation scores. For instance, in Table 2, we see an average increase of 2.8 points in citation precision, which measures the proportion of citations that actually support the claim they’re used in, and an average increase of 4.53 points in citation recall, which measures the proportion of claims that are fully supported by the included citations.
>
> We expect paragraph intents to largely impact the organization and flow of the answer, but none of the datasets we use have good automatic evaluations for this. We also note that different sub-aspects might be correlated with each other. We conjecture that using paragraph intents encourages the model to think about which of the retrieved information to use and results in increased citation scores.  We have also added this additional discussion to the paper in Section 4.2.
>
> **W3: significance test**
>
> Thank you for the suggestion! We conduct a paired t-test to test the hypothesis that +intent is better than the default version for the overall scores in our Table 2. For gemini-2.5-pro, the p-value is 0.013; For o3, the p-value is 0.072. Intent-aware SFT is much better than the baseline, and we see very low p-values. For instance, for qwen3-8b, the p-value for the hypothesis that intent-implicit SFT is better than baseline SFT is 5.5e-05. These p-values show that our results are statistically significant.
>
>
> **W4 and Q1: Overall score**
>
> The overall score is computed by performing an arithmetic macro-average over the sub-metric scores. We have updated the paper to explicitly state this. We will also update the introduction to highlight that a substantial amount of the performance gain we see is from improved attribution (an important part of long-form generation tasks, particularly in domains such as science and medicine).

---

> > ### Author Response · Authors · 2025-11-24
> > **Thank you for the review! - Part 2**
> >
> > **W5a and Q2:  Additional results**
> >
> > Thanks for your question. We did not originally include these results because the trends from o3 differ from the trends with other models, and we wanted to validate that there was no error in evaluation. We have since double-checked these results. The following tables present o3 performance on DeepScholar Bench and ResearchQA:
> >
> > | DSB | Overall | Nug. Cov. | Org.| Citation-P | Claim Cov. |
> > | :-------------------: | :-: | :-: | :-: | :-: | :-: |
> > | gemini-2.5-pro (ref.) | 55 | 49 | 64 | 53 | 54 |
> > | o3 | 46.8 | 47 | 61 | 39 | 40 |
> > | +intent (all) | 43.2 | 49 | 64 | 27 | 34 |
> > | +intent (paragraph-only) | 49.3 | 48 | 66 | 39 | 44 |
> >
> > | RQA | Rubrics |
> > | :-------------------: | :-: |
> > | o3 | 76.3 |
> > | +intent | 79.3 |
> >
> > Interestingly, during our experiments, we observed that o3 has much worse citation behavior than other frontier LLMs, especially on citation recall. From a qualitative analysis of 20 claims from o3-generated answers, we observe that for nearly 60% cases, the claims contain additional information about a paper added from o3’s own memory, going beyond the specific snippets provided from that paper in context. Adding citation intents seems to have mixed effects on this behavior, improving citation quality on AstaBench-SQA-CSV2 while dropping citation quality on DeepScholar Bench.
> >
> > Additionally, we report the performance of our SFT variants (same models as ones in Table 4, without further adaptation)  for DeepScholar Bench, as follows.
> >
> > | DSB | Overall | Nug. Cov. | Org.| Citation-P | Claim Cov. |
> > | :-------------------: | :-: | :-: | :-: | :-: | :-: |
> > | qwen3-8B | 56 | 46 | 59 | 57 | 62 |
> > | intent-explicit | 59.5 | 48 | 68 | 61 | 63 |
> > | intent-implicit | 60.3 | 45 | 68 | 63 | 65 |
> > | intent-multiview | 57.5 | 45 | 60 | 62 | 63 |
> >
> > Results show that our variants show generalizability to DSB: the best performing variant (intent-implicit) achieves better performance than the best performing large model (opus-4) in our Table 3. On the citation metrics, our SFT variants generally show better overall scores compared to o3 as well.
> >
> > **W5b and Q3: Citation Usage**
> >
> > More citations are not always better. We want to improve the quality of each citation. We analyzed the number of cited sentences and the total number of citations as shown in the table below (multi-cite denotes that a sentence has more than one in-line citation).
> > We found that the baseline and intent-aware models make a similar number of citations, and the improvements are due to citations being used correctly. We will edit the statement you mentioned in our original L372-373 with the new analysis.
> >
> > | Method | # Citing Sent | # multi-cite sent | # citations | # unique citations |
> > | :-------------------: | :-: | :-: | :-: | :-: |
> > | gemini 2.5 pro | 35.4 | 13.9 | 58.4  | 37.1 |
> > | +intent  | 35.5 | 12.0 | 58.4 | 59.9 | 38.4 |
> > | baseline SFT | 34.7 | 12.9 | 52.8  | 33.7 |
> > | intent-multiview SFT  | 35.2 | 12.9 | 58.9 | 38.9 |
> >
> > Thank you for your time and effort in reviewing our paper. We have incorporated your suggested experiments and believe they have strengthened the paper. We hope you increase your rating in light of these revisions. Please let us know if you have any further questions.
> >
> > References
> >
> > [1] Large Language Models are Zero-Shot Reasoners (https://arxiv.org/abs/2205.11916)
> >
> > [2] Chain-of-Thought Prompting Elicits Reasoning in Large Language Models (https://arxiv.org/abs/2201.11903)
> >
> > [3] ReAct: Synergizing Reasoning and Acting in Language Models (https://arxiv.org/abs/2210.03629)

---

> ### Author Response · Authors · 2025-12-03
> **Extending our Case Study to a User Study**
>
> Thanks again for the constructive comments and support. We would like to update our larger-scale user study on the impact of intents on readability as follows.
>
> Since submitting the paper, we have also begun conducting a larger-scale user study, with the goal of capturing the utility of intent-aware generation to users quantitatively. In particular, due to the nature of learning and exploration of DR tasks, we focused on whether the intents help users better decide what information to consume at both the paragraph level and at the citation level.
>
> In short, the new study takes a between-subject approach: We randomly assign participants to either a Baseline or an Experimental condition. In both conditions, participants were tasked to digest four rounds of long-form QAs, where the questions were self-proposed by these participants themselves in a pre-round of question collection. For each question, participants were given an answer report generated by gemini-2.5-pro, and would rate their self-perceived usefulness of paragraph intents (PIT) and citation intents (CIT).
>
> This new study incorporates several improvements:
>
> 1. More user-centered task design: To reduce confounds related to their prior knowledge and personal interest, participants now read reports generated on *their own questions*. They are instructed to pose/select questions that they (1) genuinely want answered and (2) do not already know the answer to.
>
> 2. Broader participant pool: We recruited more participants in total – all with a master’s or PhD degree in CS – through Prolific and university mailing lists. While we are still waiting for some results to come in (since our two-stage experiment takes some time to complete), we’ve already collected complete data from 12 participants, which has shown interesting patterns.
>
>
> In total, we collected labels from 12 participants from 42 reports, with labels for 198 unique paragraphs and 243 unique citations. On average, the participants who read with our systems report 4.35 (+-0.9) and 4.42 (+- 0.89) for paragraph and citation questions, respectively, which suggests that participants generally agree that the intents help them decide whether to read a paragraph in detail or dive into a citation. In contrast, the participants who read with the baseline system report 3.73 (+-0.99) and 3.41 (+- 1.14), which suggests the insufficiency of section titles and snippets alone. We also observe high consistency in the findings with participants hired from different sources.
>
> We also qualitatively analyzed participants’ optional free-form reflections after they completed the task, which further confirmed that our intent-aware annotations were useful: participants in the experiment condition found the annotations helpful for guiding their reading and their attention span. For example, one participant noted, “Intents are particularly useful when the report includes many hard concepts. Intents help guide the understanding of the relations among the entities”. Another annotator reported that “intent labels (BACKGROUND, USES, MOTIVATION, etc.)” can “let me quickly judge whether the citation was central to the argument or just providing broader context.”, highlighting the usefulness of the schema design. In contrast, participants in the baseline condition found the information overwhelming but still insufficient: “ the citation snippet is hard to read and understand the relevance when they are long”.
>
> We will include all details and discussion in our draft after collecting all rounds of results.

---

### Official Review · Reviewer_Rq6k · 2025-10-31

**Soundness:** 3
**Presentation:** 3
**Contribution:** 3
**Rating:** 6
**Confidence:** 4

**Summary:**

This paper augments the training data for language models with intents on paragraph and citation levels. The citation intents originate from ACL-ARC while the paragraph intents originate from existing literature. The paper is grounded in previous literature in writing intentions. The experiments include both prompting approaches with commercial closed models as well as training for open-weight models, and found improvements across different scientific writing benchmarks. Furthermore, citation scores generally improves with more citation coverage.

**Strengths:**

The paper presents an original approach to improving the citation quality by explicitly prompting and training models to generate the paragraph and citation intentions.
The motivation is well-grounded in previous literature in writing intentions, and show clear gains on several types of models and benchmarks.
The approach can be useful for the community as citation and trustworthiness is a critical flaw in modern large language models.
Finally, the paper is overall well-written with clarity, and the figures are well designed and easy to understand.

**Weaknesses:**

The paper could benefit from additional validation on the paragraph intent. Although the human studies in 4.3 shows that humans may find the intention useful for when reading and understanding the paragraph, it’s unclear if models are generating the correct intent type.

Furthermore, additional discussion on how different distribution of paragraph intent types lead to differences in results could be insightful.

Finally, there are no ablations on how much citation intent helps versus paragraph intents help, so ablating each component would improve the understanding on the importance of these part.

Typo: Table 1 “he” → “The”

**Questions:**

When generating the citations for the training data, does the model get the cited article/text as part of the input?

Are there o3 results for DeepScholarBench and RQA?

---

> ### Author Response · Authors · 2025-11-24
> **Thank you for the review! - Part 1**
>
> Thank you for your valuable and detailed feedback. We are glad you think the approach will be useful for the community! We address your questions below.
>
> **W1, W2: Discussion on Paragraph Intent Types**
>
> As shown in the intent type study in Section 4.3 and Table 6, Gemini-2.5-Pro can generate paragraph intent types as we instructed, with less than 1% error. It is not a very hard task for the large models to distinguish if a paragraph will be about Definition or Problem-Solution, especially considering that multiple types can be correct for a paragraph.
>
> As for the relevance between the distribution of paragraph intent types and performance, we perform an additional analysis as follows: we split the query-answer pairs from Gemini-2.5-Pro on AstaBench-SQA-CSV2 (with intents) into two buckets based on the Overall performance, and then compare the distribution differences in intent types. We report the count of each type as follows
>
> | Bucket / Type | Exposition | Evaluation | Narration | Cause-Effect | Definition | Compare-Contrast | Argumentation | Problem-Solution |
> | :-------------------: | :-: | :-: | :-: | :-: | :-: |  :-: | :-: | :-: |
> | Bottom half | 289        | 23       | 18      | 14      | 34         | 28     | 23| 82 |
> | Top half | 299        | 27         | 23    | 4        | 30         | 30    | 30 | 89 |
>
> We generally see that most intent types are used in both buckets and are not strongly correlated with good/bad performance. We see some patterns with “Cause-Effect” and “Argumentation” leading to lower performance, which suggests that the model potentially struggles when writing paragraphs of these types. That being said, the sample size is a little low for us to draw strong conclusions.
>
> We also conduct a similar study for citation intents with bucketing based on the average of citation precision and recall performance:
>
> | Bucket / Type | Background | Motivation | Uses | Extension | Comparison | Future |
> | :-------------------: | :-: | :-: | :-: | :-: | :-: |  :-: |
> | Bottom half | 488        | 155        | 859  | 24        | 60      | 12     |
> |Top half | 472        | 131        | 1031 | 17        | 84     | 12     |
>
> Again, most types appear in both buckets. We observe that “Uses” and “Comparison” appear more frequently in answers with higher scores, and “Motivation” occurs less frequently in answers with higher scores.
>
> **W3: Citation vs Paragraph Intent Ablations**
>
> Thanks for the comments. We kindly refer to Table 5 in the paper, where we report the performance of Gemini-2.5-Pro on AstaBench-SQA-CSV2 when using only citation intents, only paragraph intents, and both intent types together. Both citation and paragraph intents provide boosts when used separately, but the variant using both intent types performs best. The citation intent variant mostly achieves improvements on citation metrics, as expected. For the paragraph intent variant, it is harder to measure utility because we expect it to improve the overall flow and organisation of the answer, but none of the datasets we use have good automatic evaluations for these aspects. We are currently conducting a larger-scale annotation study and hope to have additional insights on the utility of both citation and paragraph intents.
>
> **Q1: Retrieval Information in Input**
>
> The retrieved information is provided to the synthetic data generation model. As described in Appendix A.7 of the paper, a list of snippets from retrieved papers is provided to the model as section_references. Each snippet is in the format “[Citation X] Snippet.” We have edited the appendix to explicitly clarify this.

---

> > ### Author Response · Authors · 2025-11-24
> > **Thank you for the review! - Part 2**
> >
> > **Q2:  o3 results for DeepScholar Bench and RQA**
> >
> > The following tables present o3 performance on DeepScholar Bench and ResearchQA:
> >
> > | DSB | Overall | Nug. Cov. | Org.| Citation-P | Claim Cov. |
> > | :-------------------: | :-: | :-: | :-: | :-: | :-: |
> > | gemini-2.5-pro (ref.) | 55 | 49 | 64 | 53 | 54 |
> > | o3 | 46.8 | 47 | 61 | 39 | 40 |
> > | +intent (all) | 43.2 | 49 | 64 | 27 | 34 |
> > | +intent (paragraph-only) | 49.3 | 48 | 66 | 39 | 44 |
> >
> > | RQA | Rubrics |
> > | :-------------------: | :-: |
> > | o3 | 76.3 |
> > | +intent | 79.3 |
> >
> > Interestingly, during our experiments, we observed that o3 has much worse citation behavior than other frontier LLMs, especially on citation recall. From a qualitative analysis of 20 claims from o3-generated answers, we observe that for nearly 60% cases, the claims contain additional information about a paper added from o3’s own memory, going beyond the specific snippets provided from that paper in context. Adding citation intents seems to have mixed effects on this behavior, improving citation quality on AstaBench-SQA-CSV2 while dropping citation quality on DeepScholar Bench. We did not initially include results from o3 because the trends were different from other models, and we wanted to double-check the results to make sure there were no errors.
> >
> > Additionally, we report the performance of our SFT variants (same models as ones in Table 4, without further training)  for DeepScholar Bench, as follows.
> >
> > | DSB | Overall | Nug. Cov. | Org.| Citation-P | Claim Cov. |
> > | :-------------------: | :-: | :-: | :-: | :-: | :-: |
> > | qwen3-8B | 56 | 46 | 59 | 57 | 62 |
> > | intent-explicit | 59.5 | 48 | 68 | 61 | 63 |
> > | intent-implicit | 60.3 | 45 | 68 | 63 | 65 |
> > | intent-multiview | 57.5 | 45 | 60 | 62 | 63 |
> >
> > Results show that our models generalize to DeepScholar Bench: the best performing variant (intent-implicit) achieves better performance than the best performing large model (opus-4) in our Table 3. On the citation metrics, our SFT variants generally show better overall scores compared to o3 as well.
> >
> > We have also fixed the typo you pointed out. Thank you for your time and effort in reviewing our paper. We have incorporated your suggested experiments and believe they have strengthened the paper. We hope you increase your rating in light of these revisions. Please let us know if you have any further questions.

---

> > > ### Comment · Reviewer_Rq6k · 2025-11-27
> > >
> > > Hi authors, thanks for the detailed response and additional results. I believe the paper would benefit from the additional discussion and experiments. I will keep my positive score.

---

> > > > ### Author Response · Authors · 2025-11-27
> > > > **Thank you for your reply!**
> > > >
> > > > Dear Reviewer Rq6k,
> > > >
> > > > Thank you for your thoughtful review and positive scores! We are glad that you like our responses, additional results, and the improved manuscript. We truly appreciate your constructive feedback and support.
> > > >
> > > > Best Regards,
> > > > Authors

---

### Official Review · Reviewer_N4jP · 2025-11-01

**Soundness:** 3
**Presentation:** 3
**Contribution:** 3
**Rating:** 6
**Confidence:** 4

**Summary:**

The paper tackles long-form, citation-heavy scientific report generation by large language models (LLMs). The authors argue that while current “deep research” systems can retrieve dozens or hundreds of sources and stitch together long answers, they still struggle with (i) organizing material into a coherent narrative, (ii) making clear rhetorical moves (e.g., motivate, compare, explain causality), and (iii) citing sources in a disciplined, honest way.

The core hypothesis is: if we make models explicitly consider their writing and citation intent while generating, those outputs will improve in structure, attribution quality, and readability

**Strengths:**

The idea of incorporating citation intention into the prompts and training data of research agents is spot-on. The argument for a need of intention in writing and research citations is well-established. The experiments are designed to thoroughly test the potential of intent-aware generation (isolating effects of prompt-based inference techniques and finetuning strategies).

**Weaknesses:**

1. **Unclear and (potentially) unreproducible training data**: Section 3.4 mentions that training data comes from a "larger teacher model". There lacks disclosure on details of the choice and ablation of the teacher model. How exactly is the synthetic data generated? Does the teacher model ever hallucinates tags? Without these details, the paper currently suffer from reproducibility and credibility issues.

2. **Experiment results analysis**: Each evaluation metric reflects a different component of citation quality -- but there's a lack of discussion on how intent-aware inference and training improves individual components, and what it means to the qualitative citation quality. Table 2 shows very marginal improvement on the SQA-CS-V2 task.

3. **Limited participants in case study**: There's only 3 participants who contributed to the case study in Section 4.3. This is an extremely small sample size that makes the result analysis weak.

**Questions:**

1. Benchmarks like AstaBench and DeepScholar-Bench use LLM-based rubric scoring and automatic verifiability checks. What LLM judges are chosen? Is the judge model from a different family than the synthetic data generation model?

2. How do you explain the difference in the impact of intent-awareness between eval metrics? For example, in Table 2 and Table 5, "Rubrics" metric and "Answer P" metric almost stayed the same, where the aggregated advantage of intent-awareness seems to only come from CitationP and CitationQ. How should we interpret this?

3. How did you ensure that human participants are not biased? How did you recruit human participants?

4. Why did you choose temperature 1.0 for inference? This can lead to unstable performance especially if you only run each evaluation metrics once. Can you please show us the mean and std of Table 4 across multiple different seeds?

---

> ### Author Response · Authors · 2025-11-24
> **Thank you for the review! - Part 1**
>
> Thank you for your valuable and detailed feedback. We are glad that you agree with our intuition that explicitly using intents would improve writing. We address your comments and questions below.
>
> **W1: Reproducibility of training data**
>
> When generating answers while eliciting intents at inference time, Gemini-2.5-pro was the best-performing model for the SQA-CS-V2 dataset, and it was within 2 points of the best-performing model (Claude-4-Opus) for the other datasets (see Tables 2 and 3 in the paper). We generated synthetic training data from Gemini-2.5-pro because it is an order of magnitude cheaper than Claude-4-opus and has competitive performance with Claude-4-opus. Thank you for flagging these missing details in the paper. We introduced part of these details in Section 4.1, Line 296 in the original paper. We have revised the paper and added these synthetic data generation details to Appendix A.1.
>
> We carefully craft our prompt to ensure that the teacher model only generates valid tags in the correct format. As shown in the original Table 6, for Gemini-2.5-pro, fewer than 1% of the tags have errors.
>
> Additionally, we will open-source both our training data and model checkpoints to support future research in this area.
>
> **W2 and Q2: Result Analysis: Impact of Intent-Awareness on Various Evaluation Aspects**
>
> In our experiments with state-of-the-art LLMs, we see that metrics like rubric score and answer precision, which do not consider citation quality, do not move much in Tables 2 and 3 because such LLMs are already highly capable of extracting key facts from retrieved information and ensuring that the presented information is topically relevant to the query. However, these models are not very good at attributing the claims made in the answer. Therefore, we expect the biggest impact of using citation intents to be improving models’ attribution capabilities, and we are seeing significant improvement in citation scores. For instance, in Table 2, we see an average increase of 2.8 points in citation precision, which measures the proportion of citations that actually support the claim they’re used in, and an average increase of 4.53 points in citation recall, which measures the proportion of claims that are fully supported by the included citations. We expect paragraph intents to largely impact the organisation and flow of the answer, but none of the datasets we use have good automatic evaluations for this.
>
> Finally, for less capable, smaller open models, we see that intent-aware training provides minor boosts to rubric and answer precision scores in addition to massively improving citation metrics (Table 4). We have also added this additional discussion to the paper in Section 4.2.
>
> For more qualitative insight into how intents affect the citation quality and answer organisation, we are in the process of conducting a larger user study (described later in the response), and will provide more insights as it progresses.
>
> **W3 and Q3: Details in Case Study**
>
> We would like to emphasize that the study we presented in the paper only intends to provide illustrative cases on how intent-awareness may bring various benefits to human readers, and as a result, the sample size matches similar studies in existing published work, e.g., [1][2]. Our case study participants were CS researchers from different departments (1 HCI, 1 CV, 1 NLP) from three different universities; none of them is a co-author on the paper.
>
> Since submitting the paper, we have also begun conducting a larger-scale user study, with the goal of capturing the user-centered gain quantitatively. The new study similarly takes a within-subject approach, tasking participants to read multiple Gemini-2.5-pro-generated reports. Some of these reports include CIT and PIT annotations (our experimental condition), while others do not (baseline). The new study incorporates several improvements:
> Broader participant pool: We are recruiting annotators with a master’s or PhD in computer science through Prolific to obtain more diverse and representative expertise.
> More user-centred task design: To help reduce confounds related to prior knowledge and personal interest, participants now read reports generated on *their own questions*. They are instructed to pose questions they (1) genuinely want answered and (2) do not already know the answer to.
> We will update our response with results from this new study as it progresses.
>
> **Q1: LLM Judges**
>
> For each dataset, we follow the same setup, including the choice of LLM judge, as prescribed by the paper introducing the dataset. As stated in Appendix A.1 of our original draft, the LLM judge for AstaBench-SQA-CSV2 is Gemini-2.5-Flash, for DeepScholar Bench it is GPT-4o, and for ResearchQA it is GPT-4.1-Mini. The model used to generate the synthetic data was gemini-2.5-pro, so it is from the same model family (but a different model) as the judge for AstaBench-SQA-CSV2.

---

> > ### Author Response · Authors · 2025-11-24
> > **Thank you for the review! - Part 2**
> >
> > **Q4: Temperature and variance**
> >
> > For inference, we use the default model temperature, and that is why we use 1.0 as the temperature. For evaluation, we use the recommendation prescribed by the paper introducing the dataset. To measure the impact of the default temperature, we also computed the mean and standard deviation across multiple different seeds.
> >
> > Following your suggestion, we re-ran two variants in Table 4 with three seeds {42, 11, 2025}: no training and intent-multiview SFT. The mean and standard deviation for these variants are: 82.7 (0.91) for “no training” and 87.7 (0.06) for “intent-multiview SFT”, respectively. We did not do this for every row because of the generation time and evaluation costs.
> >
> >
> > Thank you for your time and effort in reviewing our paper. We have incorporated your suggested experiments and believe they have strengthened the paper. We hope you increase your rating in light of these revisions. Please let us know if you have any further questions.
> >
> > References
> >
> > [1] SPHERE: An Evaluation Card for Human-AI Systems (https://arxiv.org/abs/2504.07971)
> >
> > [2] LitLLM: A Toolkit for Scientific Literature Review (https://arxiv.org/pdf/2402.01788)

---

> ### Author Response · Authors · 2025-12-03
> **Extending our Case Study to a User Study**
>
> Thanks again for your comments. Extending our discussion in **W3 and Q3: Details in Case Study**, since submitting the paper, we have also begun conducting a larger-scale user study, with the goal of capturing the utility of intent-aware generation to users quantitatively. In particular, due to the nature of learning and exploration of DR tasks, we focused on whether the intents help users better decide what information to consume at both the paragraph level and at the citation level.
>
> The new study takes a between-subject approach, where some participants read multiple gemini-2.5-pro-generated reports from a baseline system, and others will read reports generated from our system with intents:
>
> 1. Baseline systems present (in GUI): (1) for paragraphs, the section titles and first sentences, with the full content folded; (2) for citations, relevant snippets from the papers cited are inline in tooltips that appear when hovering over the citation.
> 2. Our system presents automatically generated PITs (before each paragraph) and CITs (before the snippet) in our experimental condition. The participants are asked to decide if (1) the displayed information helps them understand whether they want to read this section without opening up the paragraphs; (2) they feel confident that they know what they will learn if they dive into the citation, for each paragraph and highlighted citation, respectively.
>
> For each paragraph/highlighted citation, the participants provide a Likert rating on a scale of 1-5 (from Strongly Disagree to Strongly Agree).
>
> This new study incorporates several improvements:
> 1. Broader participant pool: We are recruiting participants with a master’s or PhD in computer science to obtain more diverse and representative expertise. In this round, we recruited from two sources: (1) Personal advertising: we recruited 7 participants from 5 affiliations; (2) Prolific: we recruited 5 participants. Each annotator receives compensation of 30 USD per hour. All participants are new to the task; none participated in our previous case study.
>
> 2. More user-centered task design: To reduce confounds related to their prior knowledge and personal interest, participants now read reports generated on *their own questions*. They are instructed to pose/select questions that they (1) genuinely want answered and (2) do not already know the answer to.
>
> In total, we collected labels from 12 participants from 42 reports, with labels for 198 unique paragraphs and 243 unique citations. On average, the participants who read with our systems report 4.35 (+-0.9) and 4.42 (+- 0.89) for paragraph and citation questions, respectively, which suggests that participants generally agree that the intents help them decide whether to read a paragraph in detail or dive into a citation. In contrast, the participants who read with the baseline system report 3.73 (+-0.99) and 3.41 (+- 1.14), which suggests the insufficiency of section titles, first sentences, and snippets alone.
>
> There is also high consistency in the findings with participants hired from different sources: (1) From personal advertising, participants reading with our system report 4.0 and 4.17 for paragraph and citation questions, while the scores are 3.77 and 3.29 for the baseline system; (2) From Prolific, participants report 4.75 and 4.77 for paragraph and citation questions reading reports with our system, while 3.73 and 3.6 reading with the baseline system. Consistent annotation results from different demographics strengthen the claim in our original case study that the intent annotations in reading interfaces help support targeted comprehension.
>
> We also qualitatively analyzed participants’ optional free-form reflections after they completed the task, which further confirmed that our intent-aware annotations were useful: participants in the experiment condition found the annotations helpful for guiding their reading and their attention span. For example, one participant noted, “Intents are particularly useful when the report includes many hard concepts. Intents help guide the understanding of the relations among the entities”. Another annotator reported that “intent labels (BACKGROUND, USES, MOTIVATION, etc.)” can “let me quickly judge whether the citation was central to the argument or just providing broader context.”, highlighting the usefulness of the schema design. In contrast, participants in the baseline condition found the information overwhelming but still insufficient: “ the citation snippet is hard to read and understand the relevance when they are long”.
>
> We will include all details and discussion in our draft after collecting all rounds of results.

---

### Author Response · Authors · 2025-12-03
**Thanks for the rebuttal period!**

We are thankful to the reviewers for the insightful and constructive feedback during the rebuttal period.

We appreciate that the reviewers acknowledge that:

1. Our idea of training models to perform intent-aware generation is original and well-motivated (Reviewer N4jP, Rq6k, yQe7)


2. Experiments and analyses are well-designed and show clear gains from using the proposed intent-aware generation method on a wide range of models and benchmarks (Reviewer N4jP, Rq6k, yUc4, yQe7)

3. Our approach of generating intents can improve model trustworthiness (Reviewer Rq6k)

4. The paper is well-written and easy to understand (Reviewer Rq6k, yUc4)

During the discussion period, we have addressed and responded to concerns and questions raised by the reviewers. Below, we summarize the main concerns:

1. In response to Reviewer N4jP, we first provide additional details about the training data, experiments, case study (and a follow-up user study), and evaluation, extending the discussion in Section 4.1 of our original draft. We also clarified that we plan to open-source both our training data and model checkpoints to support future research in this area. We also performed an additional run to demonstrate that our method has low variance and is better than the baselines.

2. In response to Reviewer Rq6k, we first provide additional analyses on the relationship between intent types and model performance, extending Section 4.3 of our original draft. We then present additional results from applying our methods to more LLMs on DSB and RQA that continue to show performance gains, and validate the generalizability of our SFT variants on DSB, with our performance matching the best-performing commercial large models. These additional results were acknowledged by the reviewer as beneficial for the paper.

3. In response to Reviewer yUc4, we first provide additional comparisons with other inference-time scaling methods (specifically CoT and ReAct baselines) and show that our method is the most performant. Then, we add paired t-tests and analyze citation usage to further establish the effectiveness of our method. We also show additional results from applying our methods to more LLMs on DSB and RQA that continue to show performance gains, and validate the generalizability of our SFT variants on DSB. We believe our new results address all major concerns raised by the reviewer.

4. In response to Reviewer yQe7, we first discuss how our method conceptually differs from other inference-time scaling methods like the chain of thought. We also present experimental comparisons with CoT and ReAct baselines, demonstrating better performance gains with our method. Then, we conduct further ablations using more flexible intent schemas that show the robustness of our approach across various schema designs. Finally, we extend our discussion in Section 5 to briefly touch on extending our framework to less-structured domains.

Accordingly, in light of the suggestions, we improved our draft to incorporate the discussion, which is summarized as follows:

1. We add additional details, results, and discussion, e.g., significance test, for the inference time technique and data generation in Section 4.2 and Appendix A.1.

2. We extend our experiments: (1) CoT and ReAct baselines in Table 5; (2) the DSB performance with our SFT variants in Appendix A.8; (3) the flexibility of our intent-schema design in Appendix A.9.

3. We fix the typos in the draft and improve our description of the prompts used in Appendix A.7

As acknowledged by all reviewers, our method, which augments attributed long-form report generation with the intents behind writing, is well-established. We hope this simple yet effective design for both inference and training stages can inspire further research in the community. We believe our responses and edits address most of the reviewers' concerns, and our data and checkpoints will support future research in this direction.

We sincerely appreciate the area chair's effort in hosting the discussion period and the reviewers' collaborative effort. We hope that our contributions and responses can be taken into full consideration.

---

### Meta-Review · Area_Chair_5Zs3 · 2026-01-09

**Summary:**

All three reviewers converged on a broadly positive assessment (all ratings 6), viewing the paper as a valuable and practical contribution to attributed long-form QA: the proposed structured, tag-based paragraph- and citation-intent elicitation is intuitive, well written, and shows consistent improvements across models and benchmarks.

The main concerns raised by reviewers were concentrated around limited novelty and insufficient analysis and validation, rather than doubts about effectiveness. Reviewer yQe7 explicitly questioned conceptual novelty, characterizing the method as “a structured form of rationale or CoT distillation,” noting the intent schema is handcrafted, and requesting stronger comparisons to CoT/rationale distillation baselines. Reviewer yUc4 similarly pointed out missing baselines (e.g., CoT prompting and CoT-based teacher training), and emphasized that the reported gains largely stem from citation precision/recall, making broader “capability/performance” claims potentially overstated without clearer explanation, overall-score definition, and statistical testing.  Reviewer Rq6k requested stronger validation that models generate correct paragraph intents, deeper discussion of how intent distributions relate to outcomes, and clearer ablations isolating the contributions of paragraph vs citation intents. Reviewer N4jP additionally highlighted concerns about experimental/reproducibility details (teacher model and synthetic data generation, evaluation stability) and the small scale of the initial human study.

In the rebuttal, the authors addressed most of these points with concrete additions: they added CoT and ReAct baselines (on Gemini 2.5 Pro) showing a sizable gap in favor of intent-aware inference, reported paired t-tests and clarified that “Overall” is a macro-average, expanded discussion on why gains concentrate in citation metrics, and provided further analyses on intent-type distributions and additional results/clarifications on experimental details and scope.

Overall, the reviews indicate a highly actionable and empirically supported approach with moderate (or nearly moderate) novelty, supporting an accept decision (poster).

**Reviewer Concerns:**

Concerns substantially addressed in the rebuttal

•	Missing / weak baselines (CoT-style prompting):
Reviewers (notably yQe7 and yUc4) asked for stronger CoT-style baselines to justify that the gains come from intent-structured reasoning rather than generic rationale prompting. The rebuttal added zero-shot CoT and ReAct results on Gemini 2.5 Pro, showing a clear gap in favor of the intent-aware inference scheme, which directly strengthens the empirical case.

•	Clarification of “Overall” and statistical reliability; explaining where improvements come from:

yUc4 questioned how “Overall” is computed and noted that improvements appear largely driven by citation precision/recall. The rebuttal clarified that Overall is a macro-average, added paired t-tests, and expanded analysis of citation usage/metrics, making the reported gains and claims more verifiable and better grounded.

•	Ablations separating paragraph-intent vs citation-intent contributions:

Rq6k asked for clearer disentanglement of the two intent channels. The rebuttal included additional ablations (e.g., all / citation-only / paragraph-only settings on Gemini 2.5 Pro), supporting the claim that both components contribute and can be complementary.

•	Reproducibility and experimental details / stability:

N4jP raised concerns about missing details (teacher model construction, synthetic data generation, evaluation stability). The rebuttal added clarifications and additional runs indicating low variance, and stated an intent to release models/data, which improves reproducibility confidence (even if not all details are fully exhaustive).

Concerns still outstanding (or only partially addressed)

•	Limited novelty / positioning relative to rationale or CoT distillation:

yQe7’s core point—that the approach is conceptually close to a structured rationale / CoT-distillation variant and not the first to explore the general problem—remains largely a matter of positioning and conceptual contribution. The rebuttal helps by adding stronger evidence that the specific intent-tagged structure works well, but it does not fundamentally change the “incremental novelty” characterization.

Overall, the rebuttal meaningfully addressed the most actionable issues (missing baselines, metric definitions/significance, ablations, and key experimental details). The remaining concerns primarily relate to conceptual novelty which are worthwhile improvements but do not undermine the paper’s practicality and empirical effectiveness at the poster-accept level.

**Reviewer Scores:**

All reviewers were already aligned in their overall assessment (each assigned a 6), and the rebuttal primarily clarified scope, added supporting experiments, and improved analysis/description rather than changing the fundamental strengths or limitations identified in the initial reviews. As a result, I expect that each reviewer would have kept their original score unchanged, even if their confidence might increase.

---

### Decision · Program_Chairs · 2026-01-26

Accept (Poster)